# Epidemiological drivers of transmissibility and severity of SARS-CoV-2 in England

Pablo N. Perez-Guzman [1,7], Edward Knock[1,7], Natsuko Imai [1],
Thomas Rawson [1], Cosmo Nazzareno Santoni[1], Joana Alcada[2,3],
Lilith K. Whittles [1], Divya Thekke Kanapram[1,4], Raphael Sonabend[1],
Katy A. M. Gaythorpe [1], Wes Hinsley[1], Richard G. FitzJohn[1], Erik Volz [1],
Robert Verity [1], Neil M. Ferguson [1], Anne Cori[1] & Marc Baguelin [1,5,6] ✉

As the SARS-CoV-2 pandemic progressed, distinct variants emerged and dominated in England. These variants, Wildtype, Alpha, Delta, and Omicron were characterized by variations in transmissibility and severity. We used a robust mathematical model and Bayesian inference framework to analyse epidemiological surveillance data from England. We quantified the impact of non-pharmaceutical interventions (NPIs), therapeutics, and vaccination on virus transmission and severity. Each successive variant had a higher intrinsic transmissibility. Omicron (BA.1) had the highest basic reproduction number at 8.4 (95% credible interval (CrI) 7.8-9.1). Varying levels of NPIs were crucial in controlling virus transmission until population immunity accumulated. Immune escape properties of Omicron decreased effective levels of immunity in the population by a third. Furthermore, in contrast to previous studies, we found Alpha had the highest basic infection fatality ratio (3.0%, 95% CrI 2.8-3.2), followed by Delta (2.1%, 95% CrI 1.9–2.4), Wildtype (1.2%, 95% CrI 1.1–1.2), and Omicron (0.7%, 95% CrI 0.6-0.8). Our findings highlight the importance of continued surveillance. Long-term strategies for monitoring and maintaining effective immunity against SARS-CoV-2 are critical to inform the role of NPIs to effectively manage future variants with potentially higher intrinsic transmissibility and severe outcomes.

The COVID-19 pandemic has driven unprecedented surges in excess mortality[1], and caused severe disruptions to healthcare systems and economies globally[2,3]. The Omicron sub-variants are more transmissible and thus more difficult to control than ancestral lineages[4,5]. Furthermore their severity levels, although lower, remain a threat to populations and health systems[6].

The global public health response to SARS-CoV-2 has included varying degrees of non-pharmaceutical interventions (NPIs), improvement of clinical care, and vaccination[7]. The SARS-CoV-2 virus has evolved over time, with distinct variants of concern (VOCs) predominant during successive epidemic waves. Epidemiological studies have assessed changes in the transmissibility and severity of SARS-

[1]MRC Centre for Global Infectious Disease Analysis and Abdul Latif Jameel Institute for Disease and Emergency Analytics (J-IDEA), School of Public Health, Imperial College London, London, UK. [2]Adult Intensive Care Unit, Royal Brompton Hospital, London, UK. [3]Department of Surgery and Cancer, Faculty of Medicine, Imperial College London, London, UK. [4]Department of Engineering, Division of Electrical Engineering, University of Cambridge, Cambridge, UK. [5]National Institute for Health Research (NIHR) Health Protection Research Unit (HPRU) in Modelling and Health Economics, London, UK. [6]Centre for Mathematical Modelling of Infectious Diseases, Department of Infectious Disease Epidemiology, London School of Hygiene & Tropical Medicine, London, UK. [7]These authors contributed equally: Pablo N. Perez-Guzman, Edward Knock. ✉e-mail: m.baguelin@imperial.ac.uk

CoV-2 and the overall effectiveness of interventions over these sequential waves of infections[8–13]. However, integrated quantitative analyses of the relative effect of different interventions and a direct comparison of the transmissibility and severity of different variants have not been performed to date.

Here we expanded a validated dynamic transmission model using a Bayesian evidence synthesis framework[8–10], and fitted it to comprehensive COVID-19 surveillance data by age and English National Health Service (NHS) regions. Data included the number of positive and negative PCR tests in the community, population-representative infection prevalence surveys, genetic characterisation of a sample of the PCR positive cases, seroprevalence from blood donor residual sera, hospital admissions and deaths, and community deaths (see Fig. 1a, b, methods and online Supplement). Leveraging these data, we estimated the absolute and relative levels of transmissibility and severity of the Wildtype, Alpha, Delta, and initial Omicron (BA.1) variants, and other key epidemiological parameters. We evaluated the relative effect of changes in contact rates due to NPIs, changes in healthcare provision and clinical practice and infection- or vaccine-induced population immunity on virus transmission and severity in England between March 16, 2020 and February 24, 2022.

## Results

### Evolving SARS-CoV-2 transmissibility and population infection control strategies

England experienced multiple COVID-19 waves, partly due to different SARS-CoV-2 variants emerging over the study period (Fig. 1a, b). The effective reproduction number, $R_t^{eff}$, quantifies viral transmissibility over time and represents the average number of secondary infections each primary infection generates at time $t$, with $R_t^{eff} < 1$ indicating epidemic decline and $R_t^{eff} > 1$ epidemic growth[14]. $R_t^{eff}$ is determined by the inherent transmission potential of a variant and by contact patterns and immunity levels in the host population[14]. We estimated both $R_t^{eff}$ and the time-varying reproduction number in the absence of population immunity, $R_t$. The latter allows us to disentangle the intrinsic and extrinsic factors that drive transmissibility, with the basic reproduction number, $R_0$, defined as $R_{t=0}$.

Viral evolution tends to select for more transmissible lineages[15]. For the Alpha and Delta variants, mutations in the spike protein that conferred higher receptor affinity and faster fusion into host cells increased intrinsic transmissibility[16]. The later Omicron (BA.1) variant had further mutations allowing substantial immune escape[16]. The $R_0$ of SARS-CoV-2 variants in England sequentially increased from 2.6 (95%

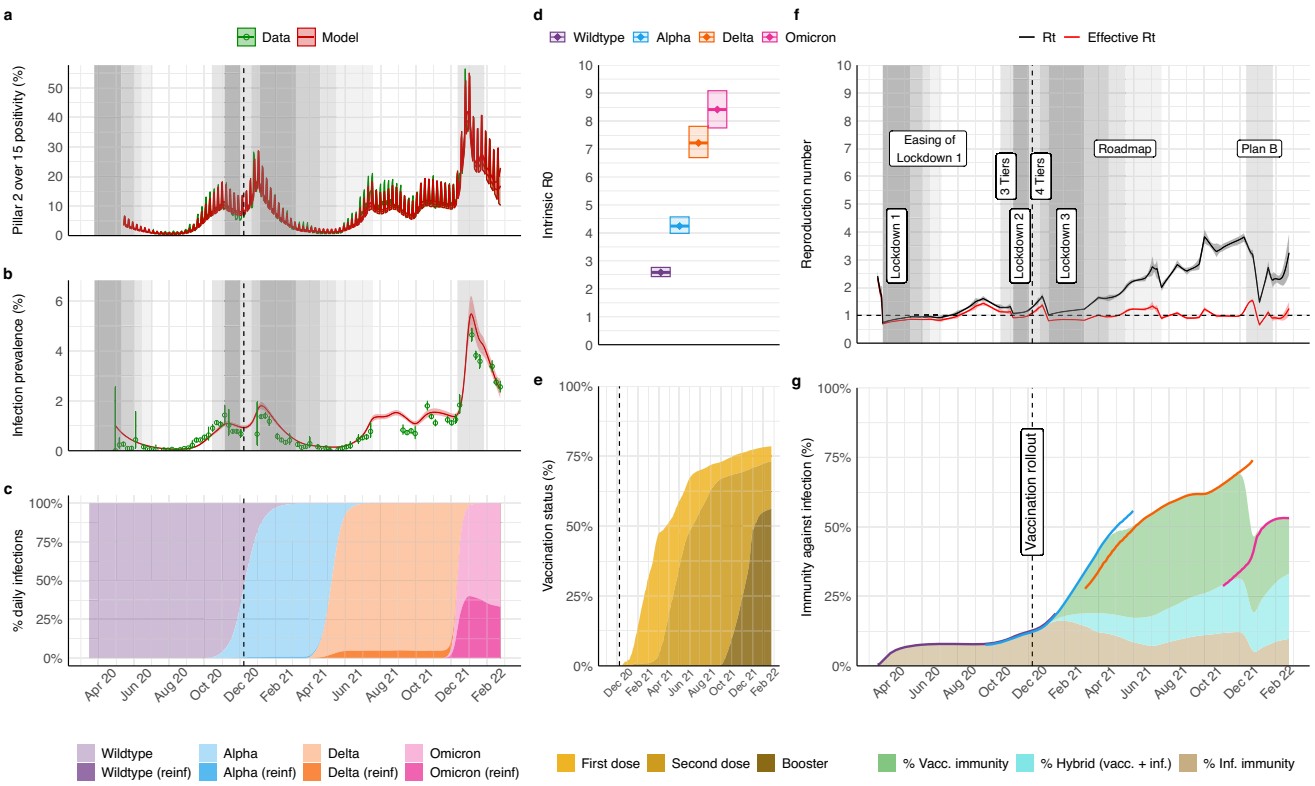

**Fig. 1 | Population-level transmission of SARS-CoV-2 between March 2020 and February 2022 in England. a** Infection positivity amongst individuals 15 years and older in the community through the national PCR testing programme (Pillar 2). **b** Infection prevalence (PCR) in representative samples of households from the REACT-1 study. Panels a and b show data in green (lines and point, with binomial 95% confidence intervals in error bars) and model trajectory in red (average and 95% credible interval (95% CrI)). Grey shading indicates periods of non-pharmaceutical interventions (NPIs) of interest for the analysis; see in **f**. For a complete list of modelled change points in contact rates see Supplement Table S11 and Figs. S28 and S29. **c** Model-inferred average frequency of daily infections by variant and type of infection (either primary or re-infection following any prior infection, "reinf"). **d** Intrinsic basic reproduction number (R0) estimates by variant (mean and 95% CrI). **e** Model trajectory of vaccine status of the national population (all ages as denominator), as informed by official data of daily doses administered (see sources in Table S1); transition between vaccination classes was modelled

stochastically (Supplement section 3.2) to capture smooth changes in population-level immunity over time. **f** Model trajectories of the instantaneous reproduction number in the absence of the effect of immunity (Rt) or accounting for immunity (effective Rt). Legends and grey areas specify date and duration of official NPIs in England over the study period. **g** Inferred effective levels of immunity in the population (all ages as denominator). Lines correspond to the effective immunity against specific variants, with colour scheme as in **c** and **d**, whereas areas indicate overall effective immunity by type of immunity (vacc: vaccine-induced, inf: from prior infection, or hybrid). Note that the coloured areas corresponding to the different types of immunity cover different periods of variant dominance and should be interpreted in the context of the circulating variants (see Supplement section 4.8). During periods of variant replacement (e.g. Alpha to Delta) the effective immunity transitions from the levels associated with the variant being replaced to the level of the variant that becomes dominant.

credible interval (CrI) 2.4–2.8) for the initial Wildtype virus, to 4.2 (95% CrI 4.0–4.6), 7.2 (95% CrI 6.7–7.8) and 8.4 (95% CrI 7.8–9.1), respectively, for the Alpha, Delta and Omicron (BA.1) variants (Fig. 1d).

NPIs were first introduced in England in mid-March 2020, during the first epidemic wave, to reduce transmission through social distancing, and limit numbers of severe cases from overwhelming the health system[17]. These measures were lifted and re-implemented thereafter, at varying levels, through to late February 2022[7]. $R_t$ varied as contact rates decreased or increased in response to NPIs (Fig. 1f). Before the COVID-19 vaccination programme started, $R_t^{eff}$ was only slightly lower than $R_t$, indicating very low levels of infection-induced immunity in the population. Accounting for waning of infection-induced immunity, we estimate that only 7.9% (95% CrI 7.4–8.3) of the population were immune by early September 2020, before the emergence of the Alpha variant (Fig. 1g).

The rollout and scale-up of the national vaccination programme, from December 2020 onwards, coincided with the dominance of more transmissible VOCs. Nevertheless, rapidly increasing immunity helped limit the spread of the Alpha and Delta variants (Fig. 1e and g). Population-level immunity against infection rapidly rose from 12.6% (95% CrI 11.9–13.2), purely from prior infections, on December 8, 2020, to 33.9% (95% CrI 32.9–35.0) by the time Delta emerged in March 2021 due to a combination of infection- and vaccine-induced immunity. We estimate population-level immunity peaked at 68.1% (95% CrI 66.3–70.1) in late November 2021, with vaccination contributing the most to this immunity profile (Fig. 1.). During this same period, in the summer of 2021, NPIs were progressively lifted under the national policy for a Roadmap out of lockdown[7,17]. $R_t$ increased gradually as NPIs were lifted in the first half of 2021, leading to a sustained high level of Delta infections for between June and October 2021 (Fig. 1a). During this period, $R_t^{eff}$ estimates remained close to 1 (Fig. 1f), suggesting increases in contact rates were balanced by increasing immunity for an extended period.

In contrast, the emergence and dominance of the initial Omicron BA.1 variant marked a decline in effective population immunity (Fig. 1g). The relative increase in Omicron's $R_0$ compared to Delta was not as high compared to previous variant replacements (Fig. 1d). Rather, the replacement of the Delta variant by Omicron was driven by the antigenic divergence and thus immune escape properties of Omicron relative to prior variants. Population-level immunity against infection decreased by a third relative to its late November peak value to reach 47.0% (95% CrI 45.4–48.6) in late December 2021 (Fig. 1g), during which period infection levels rose to an all-time high and re-infections accounted for up to a third of daily new infections (Fig. 1c). Reimplementation of some NPIs (work from home and wearing face masks and/or showing proof of negative COVID-19 tests in public venues)[7,18] introduced between December 12, 2021 and January 27, 2022, decreased $R_t$ (Fig. 1f). Nevertheless, we estimate that the combined effect of rapidly rolling out booster vaccinations and infection-induced immunity, increased population-level immunity and, consequently, decreased $R_t^{eff}$ to around 1 (Fig. 1f-g). By February 24, 2022, when all NPIs were lifted and large parts of routine surveillance were phased out[7], population-level immunity against infection was 53.2% (95% CrI 50–55.7), with 20.0% (95% CrI 19.4–20.6) resulting from vaccination, 9.7% (95% CrI 9.1–10.2) from prior infection, and 23.5% (95% CrI 21.9–24.9) from a combination of both (Fig. 1g).

### Epidemiological drivers of the severity of COVID-19 waves

The severity of SARS-CoV-2 across multiple epidemic waves in England varied substantially over the study period (Fig. 2a–c). Observable severity as traditionally measured from epidemiological surveillance, however, is subject to a range of biases, including case ascertainment, healthcare provision and population immunity[19]. Therefore, traditional surveillance cannot directly quantify the intrinsic severity of pathogens, nor disentangle the effects of underlying factors on severity.

Our model captures temporal changes in the infection hospitalisation ratio (IHR, probability of hospitalisation given infection), hospital fatality ratio (HFR, probability of death given hospitalisation for severe disease), and infection fatality ratio (IFR, probability of death given infection) by SARS-CoV-2 variant. During the time of the study, the virus ecosystem underwent many transformations, with changes taking place for example in the virus biology, the surveillance landscape and the healthcare system. To enable a like-for-like comparison of variant severity, we defined the variant-specific basic IHR, HFR and IFR as what would be observed in an entirely immunologically naïve population given baseline contact rates, and assuming the same healthcare provision that was seen after the peak of the first wave of the pandemic (see Supplement section 4.7). Lastly, we model changes in healthcare provision and changes in clinical practice throughout the duration in the study using a time varying piece-wise linear function (see Supplement Table S9). Despite the large amount of data used to inform the model, it can be challenging, if events happen simultaneously, to pinpoint which specific event affected changes in severity.

The basic IHR of SARS-CoV-2 increased from 2.2% (95% CrI 2.1–2.3) for Wildtype, to 3.4% (95% CrI 3.1–3.6) for Alpha, 4.2% (95% CrI 3.8–4.4) for Delta, but subsequently decreased to 3.1% (95% CrI 2.7–3.3) for Omicron (Fig. 2a). The overall risk of dying for hospitalised patients was highest for the Alpha variant, with a basic HFR of 50.3% (95% CrI 46.7–53.6), lower for Wildtype and Delta, at 30.0% (95% CrI 30.7–35.3) and 32.3% (95% CrI 27.4–35.8), respectively, and lowest for Omicron (BA.1), at 15.5% (95% CrI 13.1–18.8) (Fig. 2b). The age-specific IFR was a function of the age-specific IHR and HFR, further accounting for COVID-19 deaths in the community (outside hospitals). Accounting for their age-distributions (see Supplement section 4.7), the basic IFR was highest for the Alpha variant at 3.0% (95% CrI 2.8–3.2), followed by Delta at 2.1% (95% CrI 1.9–2.4), Wildtype at 1.2% (95% CrI 1.1–1.2), and Omicron at 0.7% (95% CrI 0.6–0.8) (Fig. 2c).

We also estimated the effective observable severity (given healthcare provision, immunity, and age-dependent mixing patterns) of the variants over time and estimated the effect of changes in healthcare practices and population immunity against severe outcomes. During the first epidemic wave, effective treatments, including remdesivir and dexamethasone, were approved and deployed[20,21], and there was a major re-adaptation of hospital capacity to manage severe COVID-19 cases[22]. We infer that such changes in healthcare reduced the effective HFR, in line with prior observational evidence[11], and IFR below basic levels for the then-dominant Wildtype variant (Fig. 2b, c).

Our findings also suggest that pressures on the healthcare system had a detrimental impact on the severity of the pandemic. Between October 2020 and January 2021, when a tiered geographically-localised system of NPIs in England was used[17], infection incidence increased, associated with a sustained period when $R_t^{eff} > 1$ (Fig. 1a, b and f). There was a progressive increase in all effective severity metrics, particularly in HFR (Fig. 2a–c). The model infers there was an overall higher risk of death in hospital, independently of the basic severity properties of the Alpha variant (Fig. S11 and Table S22). Given there was no representative data available on hospital deaths by variant during this period to fit our model to, we cannot fully differentiate the specific contribution of variant and healthcare effects on the increased severity. However, in an additional statistical analysis using linked patient-level records, we observed that the increase in HFR during this period was positively correlated with daily critical care bed occupancy levels, with variation across English regions (see Supplement section 6.1).

Vaccination decisively reduced the effective severity of COVID-19 in England. As age is a major predictor of COVID-19 severity[23], the national vaccination strategy prioritised the elderly and clinically vulnerable[24]. Our model reproduces this prioritisation and the complex observed dynamics of the age distribution of hospitalisations and deaths over time (see Supplement Fig. S11). We were, thus, able to infer the age-specific severity profile (model parameters were estimated as

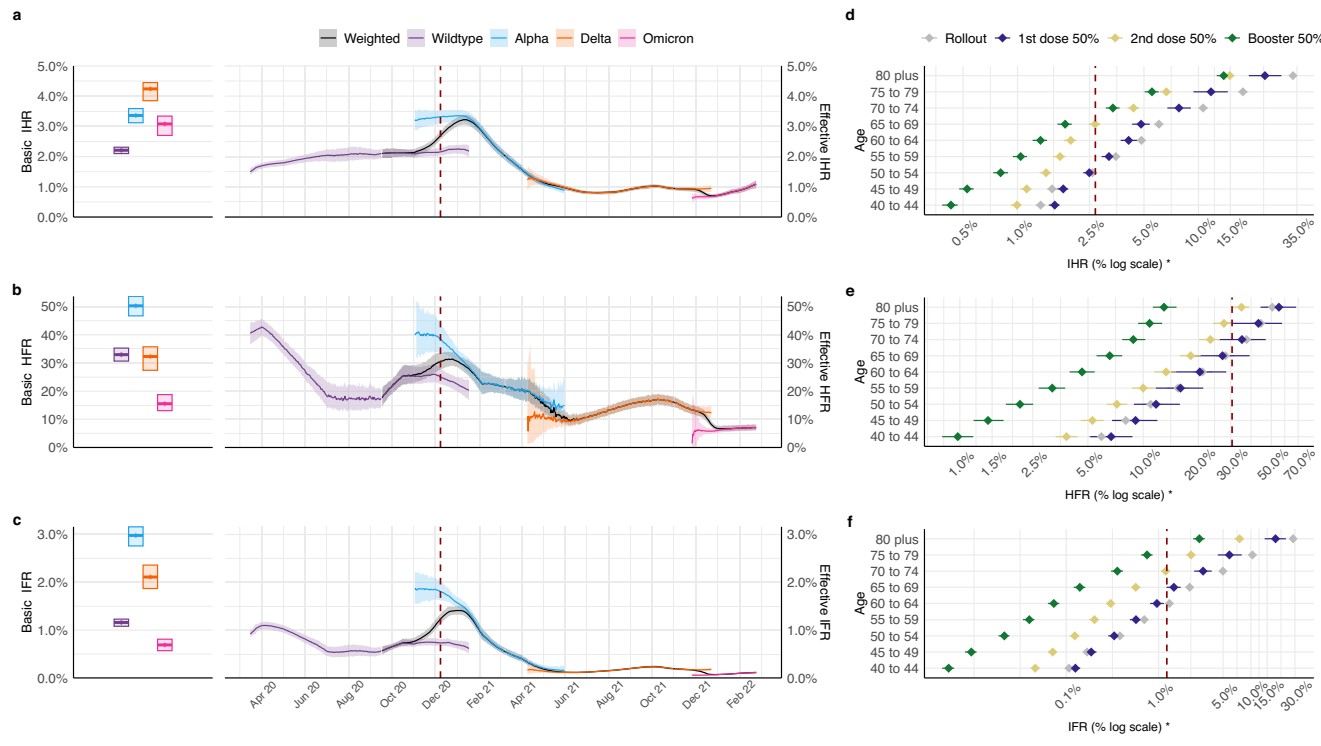

Red dashed lines: date of vaccination rollout in effective severity plots (A–C), and national effective severity (weighted) at vaccination rollout in severity by age plots (D–F).

*Effective severity by age at the time vaccination dose coverage was 50% of the entire national population.

**Fig. 2 | Inferred severity of SARS-CoV-2 variants in England between March 2020 and February 2022.** a–c Inferred basic and effective infection hospitalisation ratio (IHR, a), hospital fatality ratio (HFR, **b**) and infection fatality ratio (IFR, **c**). Boxplots to the left show the basic severity measure and trajectories to the right show severity measure over time (mean and 95% credible interval, 95% CrI) for each variant, with black showing the weighted average across co-circulating variants at any time. Basic severity is measured for each variant assuming healthcare

characteristics of the early epidemic, to allow a like-for-like comparison. Effective severity trajectories account for changing vaccine- and infection-induced protection against severe disease, as well as underlying healthcare variations (see Table S8 for model time-varying severity parameters). **d–f** Age-specific (selected age groups) effective IHR (**d**), HFR (**e**) and IFR (**f**), assessed at the date of key milestones of the national vaccination programme, "rollout" refers to the start of the vaccination programme on 8th December 2020.

in calibration or informed by the literature, see Supplement Table S11). We ran sensitivity analyses for parameters affecting the transmissibility of all the variants and, for the case of Omicron only, the cross-protection conferred from a prior infection by any variant and the vaccine effectiveness of booster doses (see Supplement section 5). We found that qualitatively our estimates were robust to assumptions about the serial interval and effectiveness of booster vaccines. Similarly, varying assumptions about the cross-protection that infection with previous variants provide against Omicron yielded very similar results to our main analysis[25].

## Discussion

Many factors influence pathogen transmission and severity, including pathogen evolution[15], intervention-based or self-adopted changes in behaviours[26], and changing infection- and/or vaccine-induced immunity[19]. Estimating the relative role of these factors in shaping an epidemic is difficult, as multiple complex interacting processes are involved, which are not directly measurable through surveillance[19]. By retrospectively fitting a dynamic transmission model to the uniquely rich COVID-19 epidemiological surveillance data from England[27] we provide the first quantitative estimates of the population-level drivers of SARS-CoV-2 transmission and severity. Although our study focuses on England, the results have implications for the effectiveness of control measures globally in this and future pandemics.

Prior to the availability of effective vaccines, at a time when population immunity was low, only stringent levels of NPIs were effective in limiting the transmission of SARS-CoV-2 in England. The intrinsic transmission advantage of Alpha and Delta allowed them to

become dominant, while Omicron's additional immune escape properties, which reduced population-level immunity against infection by approximately a third, enabled it to become dominant from late 2021 despite the high population immunity at that time. Our approach allowed us to explicitly explore the population-level role of imperfect cross-protection conferred by prior infection by historic vs emerging SARS-CoV-2 variants. We were unable to account for biological nuances underpinning cross-protection, such as the individual-level production of neutralising antibodies[28], or a different rate of infection-induced immunity waning against infection *vs* against hospitalisation or death[29]. Our results, however, were robust to variations in specific cross-protection parameters used across, included outcomes such as infection, hospitalisation or death (see sensitivity analysis in Supplement section 5).

We found that the Alpha variant had the highest basic HFR and IFR, followed in decreasing order by the Delta, Wildtype and Omicron variants. A higher HFR of Alpha compared to Delta has indeed been previously suggested by an observational study[30]. Our findings, however, contrast with the higher risk of hospitalisation and death estimated for Delta compared to Alpha in a previous meta-analysis[31]. The strongest evidence for this comes from cohort studies concluding a higher risk of hospitalisation and death with Delta than with Alpha infection, by comparing severity outcomes amongst unvaccinated individuals diagnosed by PCR in the community with linked hospitalisation and death records[32,33]. Notably, these studies adjusted their survival modelled outcomes for a number of sociodemographic variables, vaccination status and week of infection occurrence[32,33]. However, observational studies such as these[31–33] were restricted to short

time periods when both variants co-circulated and, as a result, are unable to capture variant-driven severity characteristics over the entire period of variant circulation. As a result, their estimates will be subject to biases inherent to the profile of patients seeking a PCR test in the population during the early phase of Delta emergence, which would not be representative of the profile of PCR-seeking behaviours across the whole period of Alpha and Delta transmission. Additionally, the period of Delta dominance was characterised by a partial replacement of the original Delta variant by the AY.4.2 sub-variant[34], which showed a decreased severity[35]. Whilst we did not explicitly model Delta AY.4.2, as it never completely replaced the original Delta variant in England[34], our inferred severity estimates for Delta include both periods of original Delta and AY.4.2 variant circulation.

In contrast, we average variant specific estimates over a longer period, integrating them dynamically with other ecological factors, including infection prevalence, PCR cases positivity and variant frequency in the latter. By integrating multiple data sources in a comprehensive evidence synthesis framework, we disentangle and reproduce the multiple drivers of severity and estimate that the IFR for Alpha was higher than for Delta. It should, nevertheless, be noted we did not have data available on hospital admissions or deaths by variant infection status at a level representative of the whole population. This means our model could have potentially weight higher, at least partly, the severity of the Alpha variant if factors we could not account for drove significant changes in severity and occurred synchronically between Alpha's emergence and before it became dominant; namely, a significant degradation of healthcare due to hospital pressures with subsequent improvement due to capacity scale-up, between mid-September and early November 2020. Taken together, findings from previous survival studies indicating a higher severity of Delta than Alpha[31–33], and those from our study clearly highlight the importance of considering representative genomic surveillance across the spectrum of pandemic disease severity. Future modelling studies, which follow an approach like ours to estimate severity over time and across pathogen transitions, would benefit from surveillance data with distinction of infections, hospitalisations and deaths by pathogen strain or variant.

In line with prior evidence[11], our results show that a rapid identification of effective pharmacological treatments[20,21] and re-adaptation of hospital capacity[22] were effective in countering the intrinsic severity of the Wildtype virus. However, we and other authors[36,37] also find strong evidence that improvements in clinical care can be nullified unless paired with effective and timely interventions to control infection rates in the population. We find that NPIs were not always effective or timely enough to control the virus. This led to a peak in hospitalisations over the winter of 2020/21 and, in turn, an increase in HFR which could not be explained solely by the severity characteristics of the Wildtype and Alpha variants, nor changes in the age distribution of hospital admissions. Lastly, we have demonstrated the critical role of risk-prioritised vaccination programmes[24] in reducing severity alongside transmission. Levels of immunity from prior infection in England remained low throughout the study period, with the combined proportion of those effectively protected by infection-induced and combined infection- and vaccine-induced immunity staying under a third of the national population. Albeit our findings hint at the crucial role of combined infection- and vaccine-induced immunity at the population level, it should be noted we did not explicitly model synergistic effects between these. Rather, we assumed each had an independent, multiplicative effect in reducing the risk of COVID-19 infections, severe outcomes (hospitalisation and death), and/or onward transmission (see details in Supplement sections 2.5 and 3.3), which we varied in sensitivity analysis (Supplement section 5).

Currently, the Omicron variant lineage dominates the pandemic landscape globally[25]. Our analysis robustly shows that the initial (BA.1) sub-variant of this lineage has a basic IHR comparable to other variants.

This highlights that, albeit Omicron's basic HFR and IFR are substantially lower than other variants, the Omicron lineage of SARS-CoV-2 variants remains a public health threat. Recent analyses from China, where cumulative levels of protection against the virus in the population are considerably lower than in England, suggest that NPIs remain crucial public health interventions[6]. We have demonstrated that a failure to mitigate transmission early can lead to increased severity and pressure on health services. Further research is needed to quantify the impact of changes in healthcare pressure metrics such as staff-to-patient ratios, safe bed occupancy, and availability of key commodities, which could be monitored for real-time epidemic analyses and integrated in future pandemic disease models.

Models can be used to systematically explore uncertainties around the target population and vaccination programmes for pandemic and seasonal respiratory pathogens[38–40]. Waning vaccine- and infection-induced immunity against Omicron subvariants is well established[4,12,41], and recent studies suggest that hybrid immunity (from prior infection and vaccination) may be more effective than repeated boosting through vaccination[42]. Our analysis around the emergence of the Omicron variant, nevertheless, demonstrates that boosting immunity by vaccination was a crucial intervention to maintain control of the transmissibility and severity of SARS-CoV-2 given viral evolution. Despite recent optimism that COVID-19 is becoming endemic[43], the risk of new SARS-CoV-2 variants emerging remains[2]. We have highlighted the public health implications of a higher viral transmissibility over time, which has resulted from both increased intrinsic transmissibility and increased immune escape properties. Even with similar severity properties to the current Omicron lineage, a variant with a higher intrinsic transmissibility will pose a significant public health threat[6]. COVID-19 has demonstrated that the role of vaccination and monitoring or estimating vaccine efficacy are critical during pandemic emergencies. How such data-intensive surveillance and analytics efforts are adapted going forward to continue monitoring immunity at a population level and inform future policy interventions such as potential large scale immunisation campaigns warrants further investigation.

As COVID-19 surveillance has been drastically scaled down[7], detecting future changes in the characteristics of SARS-CoV-2 and performing robust epidemiological analyses will be challenging. Our study highlights the importance of comprehensive quantitative frameworks to integrate evidence from multiple data streams. We demonstrate how combining data from different sources can help to identify patterns that may not be apparent (or would be wrongly attributed), such as the higher IFR estimated for Alpha than Delta in contrast to previous studies, when using a reduced number of data streams. This in turn provides important insights into the drivers of epidemic dynamics and informs options for future interventions. As the pandemic continues, optimising surveillance systems to detect significant changes in viral severity and transmissibility, trends in global case numbers, and the emergence of new VOCs is critical.

## Methods

We expanded a previously described[8–10] stochastic compartmental SARS-CoV-2 mathematical model and Bayesian evidence synthesis framework, fitting to a range of epidemiological surveillance data streams using particle Markov Chain Monte Carlo[44], Data range between March 16, 2020 and February 24, 2022 and are aggregated by England NHS (health administration) region. For a full description of the model structure, equations, parameters and fitted data, see online Supplement. Briefly, our model has an SEIR structure, stratified in 17 age classes (5-year bands from 0 to 79 and 80+). We fitted to age-specific data on PCR case positivity from the community (national Pillar 2 programme)[45], infection PCR-positive prevalence survey from the REal-time Assessment of Community Transmission (REACT) Study[46], hospital admissions, and community and hospital deaths, and

daily number of first, second, and booster vaccine doses[45], and to regionally aggregated data on daily general and ICU bed occupancy[45], and infection-PCR-positive prevalence from the Office for National Statistics[47].

To model heterogenous contact rates among age groups, we use the POLYMOD contact matrix for England[48]. We then fit a piecewise linear, time-varying multiplier, $\beta(t)$ to account for changes in transmission given NPIs or other official events, such as school holidays (Supplement Table S10). We use age-splines for the probabilities of severe disease (requiring hospitalisation) conditional on infection, need for ICU care given hospitalisation and death in hospital compartments (general bed, ICU, step-down care after ICU), as previously described[8]. To account for changes in healthcare characteristics, we further fit piecewise linear, time-varying modifiers of the probabilities of hospitalisation given severe disease, ICU admission given hospitalisation, death in hospital and death in the community, with dates defined either by official approval for key pharmacological treatments for COVID-19 in England or changes in healthcare seeking behaviours (Supplement Tables S8 and S10).

Our model accommodates two circulating variants at a time over a sequence of three strain replacement phases; namely, Wildtype and Alpha, Alpha and Delta, and Delta and Omicron. Within the two-variant model phases, infected compartments are further stratified into four classes, accounting for both primary infections and reinfections by each of the circulating variants. Additionally, recovered compartments feature a fifth class, to account for those recovered from an infection from a historic variant (e.g. during the Alpha and Delta phase, those recovered from a Wildtype infection). A reinfection occurs when an individual in the recovered compartment is infected with a new variant. We assume that an infection with a variant confers perfect immunity against that variant and variants that predate it (e.g. prior infection with Omicron confers perfect immunity against Delta and Omicron). However, to model imperfect protection from prior infection against emerging variants, we fix cross-immunity parameters as informed from the literature (Supplement Table S9). To ensure the robustness of results, we varied fixed cross-immunity parameters across a plausible range in sensitivity analyses (see Supplement section 5). We assume that individuals recovered from an infection can lose all their infection-induced immunity and return to the susceptible compartment and their next potential infection would be modelled as a new primary infection rather than a reinfection.

We model vaccination in seven classes (unvaccinated, dose one-no effect, dose one-full effect, dose two, waned from dose two, booster dose, waned from booster) and we explicitly account for protection against infection, symptomatic (mild) disease, severe disease requiring hospitalisation, death, and onward transmission (infectiousness). We assume fixed degree-type protection from vaccine, informed by population-level English analyses by vaccine type and variant (Supplement Tables S3 and S4, and Figs. S2–S5). To account for different vaccines used in England over the study period, we used a multiplicative VE weighting for AstraZeneca and Pfizer/Moderna (assuming same efficacy between the latter) uptake in the different age groups to daily data on first, second and booster doses administered by NHS England region. Transitions between vaccination classes were modelled stochastically (Supplement section 3.2). This allowed us to effectively capture smooth transitions in changing population-level immunity levels over time, including from full to waned vaccine protection (i.e. on average 24 weeks after vaccination).

To robustly capture the variant-specific properties of transmissibility and severity, we fitted our model to regionally aggregated data of variant frequency from Pillar 2 cases[49] for each variant transition. To infer variant transmissibility, we fitted variant-specific parameters of their seeding date and transmission advantage relative to the variant being replaced. We assume a fixed seeding pattern for each variant and a gradual decrease in the serial interval over successive variants as informed by the literature (Supplement Table S2). This latter assumption was also tested in sensitivity analysis (Supplement section 5). For variant severity, we fitted variant-specific parameter multipliers of the age-splines of the probabilities of hospitalisation conditional on infection, of ICU admission given hospitalisation, and of death given infection. For each variant in succession, these multipliers were relative to the variant being replaced, independent of fitted time-varying healthcare severity parameters, and scaled (up or down) the age-specific severity splines as a whole (assuming no age-specific changes in severity due to the variant).

We estimated the intrinsic $R_0$ and basic severity properties of the variants parametrically. For each NHS England region, the intrinsic $R_0$ was the product of the instantaneous reproduction number at the start of the model, $R_{t=0}$, and the variant's transmission advantage relative to Wildtype. We fit the transmission advantage for each variant as a parameter relative to the variant it replaces (e.g. Delta relative to Alpha). Lastly, we defined the basic severity of the variants as what their infection hospitalisation (IHR), hospital fatality (HFR) and infection fatality ratios (IFR) would be in a population without any immunity (either from prior infections or vaccination) on the 16th of May 2020. Thus, these basic severity measures allow us to compare variant severity in the absence of changes in severity over time or in the age distribution of infections or hospitalisations, such as may be driven by vaccination or improvements in healthcare. Given the stratification of our model compartments by age, vaccine and variant classes, coupled to our fitted splines of baseline severity pathways as described above, we were able to parametrically calculate the variant specific IHR, HFR and IFR by age and vaccination class (see Supplement, section 4.3.4). We were then able to remove the effect of vaccine-induced immunity by assessing these metrics in the unvaccinated classes only. To account for the effect of the age distribution of infections or hospitalisation, we derived population-level basic IHR, IFR or HFR by weighting age specific estimates across age groups using the eigenvector (further multiplied by age-specific basic IHR in the calculation of population-level basic HFR) corresponding to the leading eigenvalue of the next-generation matrix used for ascertaining $R_t$ (see Supplement section 4.7). Since basic severity values are calculated with the same scaling effects of the fitted time-varying severity modifiers for all variants, the effect of potential modifications is standardised across variants.

### Inclusion and ethics

Ethics permission was sought for the study through the standard ethical review processes of Imperial College London (London, UK), and was approved by the university's research governance and integrity team (Imperial College Research Ethics Committee reference 21IC6945). Patient consent was not required as the research team accessed fully anonymised data only, which were collected as part of routine public health surveillance activities by the UK Government.

### Reporting summary

Further information on research design is available in the Nature Portfolio Reporting Summary linked to this article.

## Data availability

All data used to produce the results in this article are available online at GitHub.

## Code availability

All code used to produce the results in this article are available online at GitHub.

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

## Acknowledgements
We thank all colleagues at PHE and front-line health professionals who have not only driven and continue to drive the daily response to the COVID-19 epidemic in England but also provided the necessary data to inform this study. This work would not have been possible without the dedication and expertise of said colleagues and professionals. The use of pillar-2 PCR testing data, vaccination data, and the variant and mutation data was made possible thanks to PHE colleagues, and we extend our thanks to N Gent and A Charlett for facilitation and insights into these data. The use of serological data was made possible by colleagues at PHE Porton Down, Colindale, and the NHS Blood Transfusion Service. We are particularly grateful to G Amirthalingam and N Andrews for helpful discussions around these data. We thank all the REal-time Assessment of Community Transmission (REACT) Study investigators for sharing PCR prevalence data. We sincerely appreciate Dr. Lloyd Chapman for his thorough review of our preprint manuscript, supplementary materials, and code. His comments and corrections greatly contributed to the completeness, accuracy, and reproducibility of our work. We finally thank the entire Imperial College London COVID-19 response team for support and feedback throughout. The views expressed are those of the authors and not necessarily those of the UK Department of Health and Social Care, the NHS, the National Institute for Health Research (NIHR), PHE, UK Medical Research Council (MRC), UK Research and Innovation, or the EU. This work was supported jointly by the Wellcome Trust and the Department for International Development (DFID; 221350). This work was supported by the Medical Research Council (MRC) Centre for Global Infectious Disease Analysis (grant number MR/R015600/1); this award is jointly funded by the MRC and Foreign, Commonwealth and Development Office (FCDO) under the MRC/FCDO Concordat agreement, and is also part of the European and Developing Countries Clinical Trials Partnership programme supported by the EU. This work was also supported by the National Institute for Health and Care Research (NIHR) Health Protection Research Unit (HPRU) in Modelling and Health Economics, which is a partnership between the UK Health Security Agency (UKHSA), Imperial College London, and the London School of Hygiene & Tropical Medicine (grant code NIHR200908). L.K.W. is funded by the Wellcome Trust (grant number 218669/Z/19/Z). The views expressed are those of the authors and not necessarily those of the UK Department of Health and Social Care (DHSC), FCDO, EU, MRC, NIHR, UKHSA, or Wellcome Trust.

## Author contributions

P.N.P.-G., E.K., N.M.F. and M.B. conceived the study. P.N.P.-G. and E.K. performed the main analyses. P.N.P.-G. and E.K. wrote the first draft of the paper, with A.C. and M.B. providing oversight of the writing process. T.R. performed statistical analysis to derive vaccine efficacy parameters used from available literature and drafted supplementary methods on this. P.N.P.-G., E.K., A.C., T.R., C.N.S., L.K.W., R.S., D.T.K. and M.B. developed the sircovid transmission model code. J.A. contributed to formulate the analysis around the impact of medical interventions and healthcare pressures on pandemic severity. K.A.M.G. and W.H. developed the data pipeline and provided code support to summarise all epidemiological surveillance data used for this study. R.G.F., E.K., L.K.W and M.B. developed the Bayesian inference methods used for this study, including the development of the packages odin, dust and mcstate. R.G.F. and W.H. provided software and hardware support. E.V. contributed to formulate the analysis around the impact of variants on changes in viral transmissibility and severity. R.V., N.M.F., A.C. and M.B. provided technical oversight of the project and advised on methodological aspects. N.M.F., A.C., M.B., N.I. and J.A. contextualised the overarching public health messages of the study. All authors contributed to the drafting of this manuscript and approved the final version submitted for publication.

## Competing interests

P.P.G. has consulted for Pfizer. A.C. has received payment from Pfizer for teaching of mathematical modelling of infectious diseases. R.S. and N.I. are currently employed by the Wellcome Trust, however the Wellcome Trust had no role in the study design, data collection, data analysis, data interpretation or writing of the manuscript. K.A.M.G. has received honoraria from Wellcome Genome Campus for lectures and salary support from the Bill & Melinda Gates Foundation and Gavi, the Vaccine Alliance, through Imperial College London for work outside this study. L.K.W. has received consultancy payments from the Wellcome Trust. All other authors declare no competing interests.
