## [Peer Review File · Nature Communications]

Epidemiological drivers of transmissibility and severity of SARS-CoV-2 in EnglandREVIEWER COMMENTS

Reviewer #1 (Remarks to the Author):

Re: Epidemiological drivers of transmissibility and severity of SARS-CoV-2 in England

This paper uses mathematical modelling within a Bayesian inference framework to examine SARS-CoV-2 variants and the effectiveness of PHSMs in England, for 2020 to early 2022. Results include R_0 estimates (peaking at 8.1 (95% credible interval (CrI) 6.8-9.3) for Omicron BA.1) and infection fatality rates (highest – interestingly – for Alpha at 2.8%, but also confirming Delta approximately 4 times IFR of Omicron).

The paper is approachable and extremely informative. I note it provides lots of “answers” to parameter values that my group uses, and I can only assume it will be very useful to many research groups.

One general concern I had is how suitable the authors model is for attributing how much of population immunity is due to vaccination versus prior infection versus hybrid (both infected and vaccinated). After much digging in the Supplement, I found Section 3.2 and Figs S5 and S6. Here I learnt that the transition from ‘full vaccination protection’ to ‘waned vaccine protection’ was stochastic, so that the model did effectively give a smoothly reducing population level VE. But this was not clear in the paper (to me at least), with what appeared to be a transition between full and partial protection (due to waning) at 24 weeks. I recommend the authors make it clearer in the main text of the paper that 24 was not deterministic, but stochastic – with a net effect of creating a smoothly declining population level VE (i.e. what is shown in Figs S5 and S6).

My next general concern, or recommendation, builds on the above. In the Discussion, quite a bit of text was given over to findings/conclusions that were not the key focus of this paper (e.g. the role of NPIs in slowing things down and health services capacity). I would recommend a bit more space in the paper for drawing out the assumptions and key limitations of things like VE versus natural immunity, and the adequacy of the model for these key findings. Likewise, the above point on the percentage contribution of vaccines, infection, NPIs to driving the R_{eff} down – and strengths and limitations of this model (e.g. how waning of both vaccine immunity and natural infection was specified).

My third major comment regards vaccine versus natural immunity. The authors state:

Line 252-254; “The combined proportion of those effectively protected by infection-induced and combined infection- and vaccine-induced immunity has stayed under a third of the national population.”

And lines 266-27-: “Waning vaccine- and infection-induced immunity against Omicron subvariants is well established,^{4,12,36} and recent studies suggest that hybrid immunity (from prior infection and vaccination) may be more effective than repetitive boosting through vaccination.³⁷ Yet, our analysis around the emergence of the Omicron variant demonstrates that boosting immunity by vaccination remains an important intervention to tackle potential changes in the transmissibility and severity of SARS-CoV-2 as it evolves.”

These statements are reasonable up to Feb 2022. However, they are unlikely reasonable into late 2022. Why? Because waning from natural protection is less than waning from vaccine immunity (at least in the data I have seen, and we use in our modelling). This means that over time the importance of natural immunity, and hybrid immunity, increases. Yes, that really kicks in beyond Feb 2022. However, it is deducible from this current paper’s model and the consistent with the

authors' acknowledgement in the paper of the large ongoing Omicron wave into mid- and late-2022. I would recommend that the role now of hybrid immunity could or should be canvassed in the Discussion more.

Minor comments:

Line 85: "We estimate that only 7.7% (95% CrI 7.2 – 8.3) of the population were immune by early September 2020". I see on the next page you talk about this measure 'peaking', so it must include waning – can this please be made clearer on first use (as I was initially unclear).

Related, should this 'population immunity' measure be subscripted (or similar qualification) for which variant? This is explained a bit more later in paper for Omicron where it dropped due to immune escape. But greater clarity here may help the reader.

Does your modelling require specifying hybrid immunity as a function of some combination of vaccine and natural protection? E.g. hybrid protection = $1 - (1-VE)(1-\text{natural immunity})$. Or do you bypass this to directly estimate R_t^{eff} ? If the latter, you still need some assumptions of the relativities of VE and natural immunity protection (and waning) to back out the relative contributions of NPIs, VE and natural protection. I searched the Methods and Supplementary material, but could not find anything on how hybrid immunity was specified; perhaps a brief 'conceptual' statement in the main text as to how this all works may help the reader, and some details in the Supplement?

(Fig 1 conveys so much information – very good, and very useful.)

Lines 152-154. "with the same age distribution of infections and healthcare provision that was seen after the peak of the first wave of the pandemic (see Supplement section 4.7)." Age structure matters ++++. To assist the reader in the main text, could this clause be amended to something like: "with the same age distribution of infections and healthcare provision that was seen after the peak of the first wave of the pandemic (see Supplement section 4.7; infections median age XX.X, IQR YY.Y to ZZ.Z and hospitalisations median age XX.X, IQR YY.Y to ZZ.Z)."

Line 163: "The basic IFR was a function of the basic IHR and HFR, further accounting for COVID-19 deaths in the community (outside hospitals)." And needs adjustment for different age distributions of your IFR and HFR estimates, surely. I assume the 'math' was done by strata of age. So should this amended sentence be better and more accurate? "The age-specific IFRs were a function of the age-specific IHR and HFR, further accounting for COVID-19 deaths in the community (outside hospitals)."

Line 164 to 166: "The basic IFR was highest for the Alpha variant at 2.8% (95% CrI 2.3 – 3.2), followed by Delta at 2.0% (95% CrI 1.5 – 2.4), Wildtype at 1.2% (95% CrI 1.0 – 1.3), and Omicron at 0.6% (95% CrI 0.4 – 0.8) (Figure 2C)." In my experience, readers are going to say (if only to themselves) that the CrI for Alpha and Delta overlap, so we cannot be sure that Alpha IFR was actually higher. HOWEVER, there will be many input parameters to these estimates for Alpha and Delta that were the same or highly correlated in each Monte Carlo simulation. Meaning the CrI about the DIFFERENCE

in IFR will be narrower than many readers expect. Therefore, can I recommend an amendment such as:

“The basic IFR was highest for the Alpha variant at 2.8% (95% CrI 2.3 – 3.2), followed by Delta at 2.0% (95% CrI 1.5 – 2.4), Wildtype at 1.2% (95% CrI 1.0 – 1.3), and Omicron at 0.6% (95% CrI 0.4 – 0.8) (Figure 2C). Directly comparing Alpha and Delta, the IFR risk difference was 0.8% (95% CrI X.X – Y.Y).” I suspect X.X will be about 0.4% and Y.Y about 1.2%.

Lines 175-79: “Our findings also suggest that pressures on the healthcare system had a detrimental impact on the severity of the pandemic. Between October 2020 and January 2021, There was a progressive increase in all effective severity metrics, particularly in HFR, (Figure 2A-C) as the model infers there was an overall higher risk of death in hospital, independently of the basic severity properties of the Alpha variant.” Is this a correct interpretation? When I look at figure 2B (for HFR) I see a flat line for the HFR for wildtype and alpha (separately) over this time. Put another way, the increasing HFR is a function of wildtype being replaced by alpha. Yes? (This comment also pertains to parts of the Discussion , e.g. lines 248 to 250.)

SUPPLEMENT

The model is a “discrete-time stochastic compartmental model”. I could not find the cycle length – is it hourly, daily, other?

p.5. “In contrast, the effective reproduction number, $R_{j,eff,t}$, for variant j at time t is the number of secondary infections in the actual population, accounting for immunity (natural and vaccine-induced) AND NPI POLICIES present at that time in the population.” Shouldn’t the CAPS added by me be added to this sentence?

Tony Blakely

Reviewer #2 (Remarks to the Author):

The article "Epidemiological drivers of transmissibility and severity of SARS-CoV-2 in England" by Perez-Guzman et al. presents an in-depth analysis of the epidemiological parameters associated with different stages of the COVID-19 pandemic in England. The authors used a wide range of data sources and employed a complex mathematical model and a robust statistical framework to estimate several key parameters such as the basic reproduction number (R_0), infection hospitalization rate (IHR), infection fatality rate (IFR), and hospitalization fatality rate (HFR) for different COVID-19 variants that emerged in England.

Overall, this study provides important insights into the impact of various COVID-19 variants and highlights the importance of ongoing surveillance and monitoring of the pandemic to inform public health strategies and interventions. They found a higher IFR for the Alpha variant compared to the Delta variant. They were also able to disentangle the effects of non-pharmaceutical interventions (NPIs), immune escape, and intrinsic increase in transmissibility for the different variants of concern. They found that the Omicron variant had the higher R_0 , but the relative increase was limited with respect to the increase estimated for previous variants (due to the higher immune escape of Omicron). I found the article very interesting as it shows how the combination of high-quality data and mathematical modelling can provide a comprehensive understanding of the factors driving the spread of the virus in England. The methods developed by the authors seem fair and robust (even if I ask the author to clarify some points, see below) and the same framework was employed by the authors in previous analyses and to inform the discussion around vaccination schedule, NPIs and reopening schedule.

However, I do have a few points that might improve the quality of the paper:

- 1) A central finding in the paper is the higher IFR observed for the Alpha variant compared to the Delta VOC. This represents a novel result and in contrast with the literature. So, I think that it should be better supported. To improve this result, the analysis could include:
 - a. A sensitivity analysis on changing VE for Alpha and Delta variants to see if it affects the estimated parameters, particularly IFR/HFR, in the same way as seen in the supplementary analysis for Omicron.
 - b. In the discussion, the author mentions the role of AY.4.2 sub-variant in reducing the severity compared to non-AY.4.2 ones. It would be useful, if the authors have data (or if they have ways to separate the two periods), to estimate the different IFR when variant AY.4.2 is non-dominant/dominant to see if the estimates in this case align with literature (reporting higher IFR for Delta compared to alpha during the first-time window). This analysis could be included in the SI.
- 2) The second main finding is related to the parameters assessed for the Omicron variant. The authors discuss this outcome in the discussion section, stating "Our analysis robustly shows that the initial (BA.1) sub-variant of this lineage has basic severity properties comparable to Wildtype IHR, HFR and IFR". However, based on Figures 2A-C, it appears that Omicron has a basic IHR that is approximately 60% higher than that of the Wildtype, whereas its basic HFR and IFR are lower by approximately 70% and 60%, respectively. Therefore, I find this sentence incorrect and I would advise the authors to revise it or clarify why they consider those values for Omicron comparable to Wildtype.
- 3) On a technical note, the authors employ a two-step process for model fitting that involves using a classical MCMC procedure to explore the parameter space for pre-tuning, followed by a second step with pMCMC using 192 particles (as described in section 1.3 of the SI). I would like to understand the benefits of the pMCMC stage. Traditional MCMC is easier to implement and can be more easily reproduced by other researchers that have similar data with no need to have two steps.

Could the authors clarify the benefits of using pMCMC over the traditional MCMC alone? I understand that this could be useful for real-time responses to speed up the inference process, but since this is a retrospective study I'm not sure about the benefits.

The figures S28-32 show the priors before the traditional MCMC step and the posterior at the end of the inference. I am curious about the posterior distribution parameters of the MCMC step that are used as initial values and proposal kernel for the pMCMC. Is it possible for the authors to superimpose the posterior distribution parameters onto these figures? Could the authors provide additional information about the MCMC procedure used, such as the type of algorithm, number of iterations, whether different starting conditions were tested, and how convergence has been checked?

Additionally, I am wondering why the authors used only 192 particles to estimate such a large number of parameters. I would have expected a much larger number of given the number of parameters that they are estimating. They should also clarify if they have performed sanity checks on that (if results are similar with a higher number of particles).

Minor Comments:

- Table S4 indicates that the protection against transmission for the Omicron variant is set at 50% for individuals who have just received their third dose, which is higher than the protection against Delta variant for the same people. However, the protection against transmission provided by the second dose against Omicron is either equal (Full protection) to or lower (Waned protection) to the one against Delta. What is the rationale behind this decision?
- Ref 68 of the SI is incomplete.
- Why in figure 2A-C in the effective IHR/HFR/IFR the line of wildtype is starting just before October 2020 and not from the beginning (March 2020)?
- Table S9: in the last two rows of the table the Rationale reported is "something".

Point by point response to Reviewer comments for: “Epidemiological drivers of transmissibility and severity of SARS-CoV-2 in England”

Reviewer #1

This paper uses mathematical modelling within a Bayesian inference framework to examine SARS-CoV-2 variants and the effectiveness of PHSMs in England, for 2020 to early 2022. Results include R_0 estimates (peaking at 8.1 (95% credible interval (CrI) 6.8-9.3) for Omicron BA.1) and infection fatality rates (highest – interestingly – for Alpha at 2.8%, but also confirming Delta approximately 4 times IFR of Omicron).

The paper is approachable and extremely informative. I note it provides lots of “answers” to parameter values that my group uses, and I can only assume it will be very useful to many research groups.

We thank the reviewer for the very positive feedback and interest in our results.

One general concern I had is how suitable the authors model is for attributing how much of population immunity is due to vaccination versus prior infection versus hybrid (both infected and vaccinated). After much digging in the Supplement, I found Section 3.2 and Figs S5 and S6. Here I learnt that the transition from ‘full vaccination protection’ to ‘waned vaccine protection’ was stochastic, so that the model did effectively give a smoothly reducing population level VE. But this was not clear in the paper (to me at least), with what appeared to be a transition between full and partial protection (due to waning) at 24 weeks. I recommend the authors make it clearer in the main text of the paper that 24 was not deterministic, but stochastic – with a net effect of creating a smoothly declining population level VE (i.e. what is shown in Figs S5 and S6).

Many thanks for the recommendation. We agree with the reviewer’s suggestion and have added a clarification around this issue in the caption to Figure 1 and methods section of the main manuscript.

Caption change:

261 (all ages as denominator), as informed by official data of daily doses administered (see sources in Table S1); transition
262 between vaccination classes was modelled stochastically (Supplement section 3.2), allowing the model to capture
263 smooth changes in population-level immunity over time. F) Model trajectories of the instantaneous reproduction

Main text methods expanded:

938 England region. Transitions between vaccination classes were modelled stochastically (Supplement section 3.2). This
939 allowed us to effectively capture smooth transitions in changing population-level immunity levels over time, including
940 from full to waned vaccine protection (i.e. on average 24 weeks after vaccination).

My next general concern, or recommendation, builds on the above. In the Discussion, quite a bit of text was given over to findings/conclusions that were not the key focus of this paper (e.g. the role of NPIs in slowing things down and health services capacity). I would recommend a bit more space in the paper for drawing out the assumptions and key limitations of things like VE versus natural immunity, and the adequacy of the model for these key findings. Likewise, the above point on the percentage contribution of vaccines, infection, NPIs to driving the Reff down – and strengths and limitations of this model (e.g. how waning of both vaccine immunity and natural infection was specified).

We thank the reviewer for the suggestion. We have now expanded the discussion as suggested to appraise key limitations and corresponding strengths and/or the use of sensitivity analysis to ensure robustness in our results. We explicitly highlight the challenges of modelling cross-protection, the lack of variant-specific hospital and deaths genomic surveillance data, and of modelling hybrid immunity.

539 despite the high population immunity at that time. Our approach allowed us to explicitly explore the population-level
540 role of imperfect cross-protection conferred by prior infection by historic vs emerging SARS-CoV-2 variants. We were
541 unable to account for biological nuances underpinning cross-protection, such as the individual-level production of

551 neutralising antibodies,²⁸ or a different rate of infection-induced immunity waning against infection vs against
552 hospitalisation or death.²⁹ Our results, however, were robust to variations in specific cross-protection parameters used
553 across, included outcomes such as infection, hospitalisation or death (see sensitivity analysis in Supplement section 5).

575 multiple drivers of severity and estimate that the IFR for Alpha was higher than for Delta. It should, nevertheless, be
576 noted we did not have data available on hospital admissions or deaths by variant infection status at a level representative
577 of the whole population. This means our model could have potentially weight higher, at least partly, the severity of the
578 Alpha variant if factors we could not account for drove significant changes in severity and occurred synchronically

650 between Alpha's emergence and before it became dominant; namely, a significant degradation of healthcare due to
651 hospital pressures with subsequent improvement due to capacity scale-up, between mid-September and early November
652 2020. Taken together, findings from previous survival studies indicating a higher severity of Delta than Alpha,³¹⁻³³ and
653 those from our study clearly highlight the importance of considering representative genomic surveillance across the
654 spectrum of pandemic disease severity. Future modelling studies, which follow an approach like ours to estimate severity
655 over time and across pathogen transitions, would benefit from surveillance data with distinction of infections,
656 hospitalisations and deaths by pathogen strain or variant.

663 national population. Albeit our findings hint at the crucial role of combined infection- and vaccine-induced immunity
669 at the population level, it should be noted we did not explicitly model synergistic effects between these. Rather, we
670 assumed each had an independent, multiplicative effect in reducing the risk of COVID-19 infections, severe outcomes
671 (hospitalisation and death), and/or of onward transmission (see details in Supplement sections 2.5 and 3.3), which we
672 varied in sensitivity analysis (Supplement section 5).

*My third major comment regards vaccine versus natural immunity. The authors state:
Line 252-254; "The combined proportion of those effectively protected by infection-induced and
combined infection- and vaccine-induced immunity has stayed under a third of the national
population." And lines 266-27-: "Waning vaccine- and infection-induced immunity against Omicron
subvariants is well established,^{4,12,36} and recent studies suggest that hybrid immunity (from prior
infection and vaccination) may be more effective than repetitive boosting through vaccination.³⁷ Yet,
our analysis around the emergence of the Omicron variant demonstrates that boosting immunity by
vaccination remains an important intervention to tackle potential changes in the transmissibility
and severity of SARS-CoV-2 as it evolves."*

These statements are reasonable up to Feb 2022. However, they are unlikely reasonable into late 2022. Why? Because waning from natural protection is less than waning from vaccine immunity (at least in the data I have seen, and we use in our modelling). This means that over time the importance of natural immunity, and hybrid immunity, increases. Yes, that really kicks in beyond Feb 2022. However, it is deducible from this current paper's model and the consistent with the authors' acknowledgement in the paper of the large ongoing Omicron wave into mid- and late-2022. I would recommend that the role now of hybrid immunity could or should be canvassed in the Discussion more.

Many thanks. We agree these are very important discussion points. We have clarified in our discussion that such statements apply to the role of infection-induced and hybrid immunity during the study period, but its role in the current state of the pandemic and into the future requires further investigation. Considerations raised by the reviewer are now explicitly presented in our discussion.

705 vaccination) may be more effective than repeated boosting through vaccination.⁴² Our analysis around the emergence
706 of the Omicron variant, nevertheless, demonstrates that boosting immunity by vaccination was a crucial intervention to
707 maintain control of the transmissibility and severity of SARS-CoV-2 given viral evolution. Despite recent optimism
708 that COVID-19 is becoming endemic,⁴³ the risk of new SARS-CoV-2 variants emerging remains.² We have highlighted
709 the public health implications of a higher viral transmissibility over time, which has resulted from both increased
710 intrinsic transmissibility and increased immune escape properties. Even with similar severity properties to the current
711 Omicron lineage, a variant with a higher intrinsic transmissibility will pose a significant public health threat.⁶ COVID-
712 19 has demonstrated that the role of vaccination and monitoring or estimating vaccine efficacy are critical during
713 pandemic emergencies. How such data-intensive surveillance and analytics efforts are adapted going forward to
714 continue monitoring immunity at a population level and inform future policy interventions such as potential large scale
715 immunisation campaigns warrants further investigation.

Minor comments:

Line 85: "We estimate that only 7.7% (95% CrI 7.2 – 8.3) of the population were immune by early September 2020". I see on the next page you talk about this measure 'peaking', so it must include waning – can this please be made clearer on first use (as I was initially unclear).

We have made this clarification, thank you.

124 was only slightly lower than R_t , indicating very low levels of infection-induced immunity in the population. Accounting
125 for waning of infection-induced immunity, we estimate that only 7.8% (95% CrI 7.3 – 8.3) of the population were
126 immune by early September 2020, before the emergence of the Alpha variant (Figure 1G).

Related, should this ‘population immunity’ measure be subscripted (or similar qualification) for which variant? This is explained a bit more later in paper for Omicron where it dropped due to immune escape. But greater clarity here may help the reader.

We agree with the reviewer that the very notion of population immunity is changing with the virus evolution. We believe subscripting these measures in the main manuscript could make the results more difficult to understand for non-specialised readers. However, to clarify our meaning of “*population immunity*”, we have added a clarification in the caption to Fig 1G. We also refer the reader to Supplement section 4.8, where we elaborate on how we calculated effective population-level immunity against infection and make use of subscripting.

270 immunity (vaccine-induced, hybrid or from prior infection). Note that the coloured areas corresponding to the
271 different types of immunity cover different periods of variant dominance and should be interpreted in the context of
272 the circulating variants (see Supplement section 4.8). During periods of variant replacement (e.g. Alpha to Delta) the

4.8 Effective population-level immunity against infection

We calculate the effective population-level immunity (EPI) in the susceptible and recovered population (S and R compartments) against infection with variant j by weighting individuals by 1 minus their susceptibility to variant j relative to an unvaccinated individual in the S compartment. Thus unvaccinated individuals in the S compartment are weighted 0, and we have three further types of individuals to account for

1. susceptible and vaccinated (vaccination-derived immunity only)
2. recovered and unvaccinated (infection-derived immunity only)
3. recovered and vaccinated (both vaccination- and infection-derived immunity, i.e. "hybrid immunity")

For the latter two, note that individuals in the R compartment recovered from variants j_1 or j_2 are fully protected against infection with variant j_1 , and those recovered from variant j_2 are additionally fully protected against infection with variant j_2 .

The effective population-level immunity against infection with variant j due to vaccination-derived immunity only is

$$EPI_V^j(t) = \sum_i \sum_{k>0} \left(1 - \chi^{i,j,k}(t)\right) S^{i,k}(t). \quad (290)$$

The effective population-level immunity against infection with variant j due to infection-derived immunity only is

$$EPI_I^j(t) = \begin{cases} \sum_i \eta_{j_1} R^{i,j_H,0}(t) + R^{i,j_1,0}(t) + R^{i,j_1^{reinf},0}(t) + R^{i,j_2,0}(t) + R^{i,j_2^{reinf},0}(t) & \text{if } j = j_1, \\ \sum_i \eta_{j_2} \left(R^{i,j_H,0}(t) + R^{i,j_1,0}(t) + R^{i,j_1^{reinf},0}(t) \right) + R^{i,j_2,0}(t) + R^{i,j_2^{reinf},0}(t) & \text{if } j = j_2. \end{cases} \quad (291)$$

The effective population-level immunity against infection with variant j due to hybrid immunity is

$$EPI_H^j(t) = \begin{cases} \sum_i \sum_{k>0} \left(1 - (1 - \eta_{j_1}) \chi^{i,j,k}\right) R^{i,j_H,k}(t) + R^{i,j_1,k}(t) + R^{i,j_1^{reinf},k}(t) + R^{i,j_2,k}(t) + R^{i,j_2^{reinf},k}(t) & \text{if } j = j_1, \\ \sum_i \sum_{k>0} \left(1 - (1 - \eta_{j_2}) \chi^{i,j,k}\right) \left(R^{i,j_H,k}(t) + R^{i,j_1,k}(t) + R^{i,j_1^{reinf},k}(t) \right) + R^{i,j_2,k}(t) + R^{i,j_2^{reinf},k}(t) & \text{if } j = j_2. \end{cases} \quad (292)$$

The effective population-level immunity against infection with variant j is thus

$$EPI^j(t) = EPI_V^j(t) + EPI_I^j(t) + EPI_H^j(t). \quad (293)$$

For the overall effective population-level immunity against infection, we weight by the same weightings used for the variant-weighted reproduction number (see Equation (274)).

Does your modelling require specifying hybrid immunity as a function of some combination of vaccine and natural protection? E.g. hybrid protection = 1 - (1-VE)(1-natural immunity). Or do you bypass this to directly estimate Rteff? If the latter, you still need some assumptions of the relativities of VE and natural immunity protection (and waning) to back out the relative contributions of NPIs, VE and natural protection. I searched the Methods and Supplementary material, but could not find anything on how hybrid immunity was specified; perhaps a brief 'conceptual' statement in the main text as to how this all works may help the reader, and some details in the Supplement?

(Fig 1 conveys so much information – very good, and very useful.)

Many thanks. We agree it was not completely clear how we account for hybrid immunity in our model. In the new supplement section 4.8 (see snapshot below), we clarify this is indeed a multiplicative approach, as noted by the reviewer.

4.8 Effective population-level immunity against infection

We calculate the effective population-level immunity (EPI) in the susceptible and recovered population (S and R compartments) against infection with variant j by weighting individuals by 1 minus their susceptibility to variant j relative to an unvaccinated individual in the S compartment. Thus unvaccinated individuals in the S compartment are weighted 0, and we have three further types of individuals to account for

1. susceptible and vaccinated (vaccination-derived immunity only)
2. recovered and unvaccinated (infection-derived immunity only)
3. recovered and vaccinated (both vaccination- and infection-derived immunity, i.e. "hybrid immunity")

For the latter two, note that individuals in the R compartment recovered from variants j_1 or j_2 are fully protected against infection with variant j_1 , and those recovered from variant j_2 are additionally fully protected against infection with variant j_2 .

The effective population-level immunity against infection with variant j due to vaccination-derived immunity only is

$$EPI_V^j(t) = \sum_i \sum_{k>0} \left(1 - \chi^{i,j,k}(t)\right) S^{i,k}(t). \quad (290)$$

The effective population-level immunity against infection with variant j due to infection-derived immunity only is

$$EPI_I^j(t) = \begin{cases} \sum_i \eta_{j_1} R^{i,j_H,0}(t) + R^{i,j_1,0}(t) + R^{i,j_1^{reinf},0}(t) + R^{i,j_2,0}(t) + R^{i,j_2^{reinf},0}(t) & \text{if } j = j_1, \\ \sum_i \eta_{j_2} \left(R^{i,j_H,0}(t) + R^{i,j_1,0}(t) + R^{i,j_1^{reinf},0}(t) \right) + R^{i,j_2,0}(t) + R^{i,j_2^{reinf},0}(t) & \text{if } j = j_2. \end{cases} \quad (291)$$

The effective population-level immunity against infection with variant j due to hybrid immunity is

$$EPI_H^j(t) = \begin{cases} \sum_i \sum_{k>0} \left(1 - (1 - \eta_{j_1}) \chi^{i,j,k}\right) R^{i,j_H,k}(t) + R^{i,j_1,k}(t) + R^{i,j_1^{reinf},k}(t) + R^{i,j_2,k}(t) + R^{i,j_2^{reinf},k}(t) & \text{if } j = j_1, \\ \sum_i \sum_{k>0} \left(1 - (1 - \eta_{j_2}) \chi^{i,j_2,k}\right) \left(R^{i,j_H,k}(t) + R^{i,j_1,k}(t) + R^{i,j_1^{reinf},k}(t) \right) + R^{i,j_2,k}(t) + R^{i,j_2^{reinf},k}(t) & \text{if } j = j_2. \end{cases} \quad (292)$$

The effective population-level immunity against infection with variant j is thus

$$EPI^j(t) = EPI_V^j(t) + EPI_I^j(t) + EPI_H^j(t). \quad (293)$$

For the overall effective population-level immunity against infection, we weight by the same weightings used for the variant-weighted reproduction number (see Equation (274)).

Lines 152-154. "with the same age distribution of infections and healthcare provision that was seen after the peak of the first wave of the pandemic (see Supplement section 4.7)." Age structure matters +++. To assist the reader in the main text, could this clause be amended to something like: "with the same age distribution of infections and healthcare provision that was seen after the peak of the first wave of the pandemic (see Supplement section 4.7; infections median age XX.X, IQR YY.Y to ZZ.Z and hospitalisations median age XX.X, IQR YY.Y to ZZ.Z)."

We apologise that the wording of our initial statement around age structure was inaccurate. We have removed it and clarified in the manuscript that the calculated basic severity reflects what would be observed in an entirely susceptible population given baseline contact rates. We had correctly explained in the Methods section and Supplement section 4.7 that our calculations match the age profile used to calculate R_t . That is, we used the eigenvector of the leading eigenvalue in our next

generation matrix as weights for the age-specific basic IHR and IFR. For age-specific basic HFR, we further multiply the eigenvector by the age-specific IHR, with the age weightings varying across variants in line with changes in the age-specific IHR at the point of variant emergence.

We have added a clarification about the weighting for HFR to the Methods section, and additionally a reference to Supplement section 4.7 to guide the reader to the more detailed explanation of the basic severity calculations. Please note we are not able to extract central tendency measures of the age distribution in our model without introducing additional assumptions, as the model is stratified into 17 age bins. Any summary statistics about age distribution would need to assume, for example, uniformly distributed individual ages across bins, which would be particularly problematic for the 80+ compartment (unbounded). We would, nevertheless, like to highlight Figures S15 and S27-31, which capture our model's ability to recover empiric age distributions of infections, hospital admissions and deaths over time.

296 landscape and the healthcare system. To enable a like-for-like comparison of variant severity, we defined the variant-
297 specific basic IHR, HFR and IFR as what would be observed in an entirely immunologically naïve population given
298 baseline contact rates, and assuming the same healthcare provision that was seen after the peak of the first wave of the
299 pandemic (see Supplement section 4.7). Lastly, we model changes in healthcare provision and changes in clinical

964 IFR or HFR by weighting age specific estimates across age groups using the eigenvector further multiplied by age-
965 specific basic IHR in the calculation of population-level basic HFR corresponding to the leading eigenvalue of the
966 next-generation matrix used for ascertaining R_t (see Supplement section 4.7). Since basic severity values are

Line 163: “The basic IFR was a function of the basic IHR and HFR, further accounting for COVID-19 deaths in the community (outside hospitals).” And needs adjustment for different age distributions of your IFR and HFR estimates, surely. I assume the ‘math’ was done by strata of age. So should this amended sentence be better and more accurate? “The age-specific IFRs were a function of the age-specific IHR and HFR, further accounting for COVID-19 deaths in the community (outside hospitals).”

Many thanks for highlighting this. We have made this correction, as indeed the maths were done by age-strata and then aggregating as indicated in responses above using the eigenvector of the leading eigenvalue.

308 respectively, and lowest for Omicron (BA.1), at 14.3% (95% CrI 11.3 – 17.1) (Figure 2B). The age-specific IFR was a
309 function of the age-specific IHR and HFR, further accounting for COVID-19 deaths in the community (outside
310 hospitals). Accounting for their age-distributions (see Supplement section 4.7), the basic IFR was highest for the Alpha
311 variant at 2.9% (95% CrI 2.7 – 3.2), followed by Delta at 2.2% (95% CrI 2.0 – 2.4), Wildtype at 1.2% (95% CrI 1.1 –
312 1.2), and Omicron at 0.7% (95% CrI 0.6 – 0.8) (Figure 2C).

Line 164 to 166: “The basic IFR was highest for the Alpha variant at 2.8% (95% CrI 2.3 – 3.2), followed by Delta at 2.0% (95% CrI 1.5 – 2.4), Wildtype at 1.2% (95% CrI 1.0 – 1.3), and Omicron at 0.6% (95% CrI 0.4 – 0.8) (Figure 2C).” In my experience, readers are going to say (if only to themselves) that the CrI for Alpha and Delta overlap, so we cannot be sure that Alpha IFR was actually higher. HOWEVER, there will be many input parameters to these estimates for Alpha and Delta that were the same or highly correlated in each Monte Carlo simulation. Meaning the CrI about the DIFFERENCE in IFR will be narrower than many readers expect. Therefore, can I recommend an amendment such as: “The basic IFR was highest for the Alpha variant at 2.8% (95% CrI 2.3 – 3.2), followed by Delta at 2.0% (95% CrI 1.5 – 2.4), Wildtype at 1.2% (95% CrI 1.0 – 1.3), and Omicron at 0.6% (95% CrI 0.4 – 0.8) (Figure 2C). Directly comparing Alpha and Delta, the IFR risk difference was 0.8% (95% CrI X.X– Y.Y).” I suspect X.X will be about 0.4% and Y.Y about 1.2%.

Many thanks for highlighting this very important point and suggesting a risk difference calculation. As mentioned above, we modified the calculation of national means and CrIs for the intrinsic transmissibility and basic severity of the variants.

Previously, regional means and CrIs were calculated, and national estimates of the means/lower bounds/upper bounds were produced by taking weighted means of the regional means/lower bounds/upper bounds. Now the regional posterior samples of 1000 for each of the measures are paired in a way consistent with how all other measures are paired (by ranking regional posterior samples by cumulative incidence, and by pairing same-ranked samples across regions) to create national posterior samples for the intrinsic transmissibility and basic severity of the variants, from which we directly calculate means and CrIs.

Since the older method naturally maximised variance for each measure, the newer calculated CrIs are narrower. However, the new method is more robust, and more consistent with how other national estimates are aggregated from regional posterior samples. Furthermore, the new method makes it possible for us to calculate national estimates of the relative risks of basic severity as suggested by the reviewer. These are presented for all basic severity measures (and ratios of intrinsic transmissibility) for each combination of variants in the supplement (section 6.1). Note that, in particular, the CrIs for

the basic IFRs for Alpha and Delta no longer overlap, and the CrI for their relative risk indicates that it is significant that the Alpha basic IFR is higher than that of Delta.

Moreover, we present additional analyses to further support the robustness of our findings . Across relevant sections in our supplement (6.1 - 6.4), we articulate how these additional analyses support the robustness of our results, whilst being explicit about their limitations and what they could mean in terms of our key epidemiological messages.

6.1 Relative risk of transmissibility and severity of the variants

Tables below summarise the relative risk of transmissibility and severity comparing across all variants. We performed paired sampling of model-inferred relative transmissibility and severity parameters within pMCMC steps. We did this to ensure 95%CrI of results below reflected only uncertainty from parameter uncertainty, void of stochastic variance.

	Wildtype	Alpha	Delta	Omicron
Wildtype	-	1.63 (1.60 - 1.67)	2.73 (2.66 - 2.79)	3.24 (3.14 - 3.35)
Alpha	0.61 (0.60 - 0.62)	-	1.67 (1.62 - 1.71)	1.98 (1.91 - 2.05)
Delta	0.37 (0.36 - 0.38)	0.60 (0.58 - 0.62)	-	1.19 (1.15 - 1.22)
Omicron	0.31 (0.30 - 0.32)	0.50 (0.49 - 0.52)	0.84 (0.82 - 0.87)	-

Table S18: Relative intrinsic transmissibility risk (mean, 95%CrI) across modelled variants, calculated as R_0^j / R_0^i . Note here indices i and j refer to table row and column, respectively.

	Wildtype	Alpha	Delta	Omicron
Wildtype	-	1.48 (1.39 - 1.57)	1.91 (1.80 - 2.04)	1.56 (1.36 - 1.75)
Alpha	0.67 (0.64 - 0.72)	-	1.29 (1.20 - 1.42)	1.05 (0.92 - 1.21)
Delta	0.52 (0.49 - 0.56)	0.78 (0.70 - 0.84)	-	0.81 (0.72 - 0.89)
Omicron	0.65 (0.57 - 0.73)	0.96 (0.83 - 1.09)	1.23 (1.12 - 1.38)	-

Table S19: Relative basic IHR risk (mean, 95%CrI) across modelled variants, calculated as IHR^j / IHR^i . Note here indices i and j refer to table row and column, respectively.

	Wildtype	Alpha	Delta	Omicron
Wildtype	-	1.49 (1.40 - 1.60)	1.00 (0.88 - 1.15)	0.44 (0.34 - 0.53)
Alpha	0.67 (0.62 - 0.72)	-	0.67 (0.58 - 0.77)	0.29 (0.23 - 0.36)
Delta	1.00 (0.87 - 1.14)	1.50 (1.29 - 1.72)	-	0.44 (0.36 - 0.52)
Omicron	2.31 (1.88 - 2.92)	3.44 (2.81 - 4.27)	2.30 (1.93 - 2.77)	-

Table S20: Relative basic HFR risk (mean, 95%CrI) across modelled variants, calculated as HFR^j / HFR^i . Note here indices i and j refer to table row and column, respectively.

	Wildtype	Alpha	Delta	Omicron
Wildtype	-	2.48 (2.32 - 2.64)	1.84 (1.65 - 2.03)	0.59 (0.47 - 0.72)
Alpha	0.40 (0.38 - 0.43)	-	0.74 (0.65 - 0.84)	0.24 (0.20 - 0.29)
Delta	0.54 (0.49 - 0.61)	1.35 (1.20 - 1.53)	-	0.32 (0.26 - 0.38)
Omicron	1.70 (1.39 - 2.11)	4.21 (3.48 - 5.12)	3.13 (2.66 - 3.80)	-

Table S21: Relative basic IFR risk (mean, 95%CrI) across modelled variants, calculated as IFR^j / IFR^i . Note here indices i and j refer to table row and column, respectively.

243
249
250
251

Figure 1 – Population-level transmission of SARS-CoV-2 between March 2020 and February 2022 in England. A) Infection positivity measured amongst those aged over 15 years old in the community through the national

493
499

Figure 2 – Inferred severity of SARS-CoV-2 variants in England between March 2020 and February 2022. A-C)

6.3 Severity outputs by vaccination status

We compared modelled vs empiric hospital admissions and deaths by vaccine status over time, aggregated from national patient-level linked records of vaccination status, hospitalisations and deaths. The model performed well in capturing these trends, which we did not explicitly fit to, indicating it was able to correctly infer the effect of vaccine-derived immunity against hospitalisation and death at the population level.

It should be further noted that our model includes for time-varying patterns of severity mechanisms (section 4.4.4), as proxy of healthcare performance variations over time. Results below thus support there was an overall low risk of the model wrongly attributing differences in variant basic severity to factors driven by the evolving profiles of vaccine-derived immunity or healthcare performance.

Figure S12: Points represent data on daily admissions by vaccination status (see section 6.2) for details of data aggregation from linked patient-level line lists) and shaded areas 95%CrI of model inferred trajectories by NHS England region.

Figure S13: Points represent data on daily hospital deaths by vaccination status (see section 6.2) for details of data aggregation from linked patient-level line lists) and shaded areas 95%CrI of model inferred trajectories by NHS England region.

6.4 Model comparison to other empiric severity outputs over time

There are additional nuances around variations in patient cohorting practices within the healthcare system over time we could not account for in our model. These include changes in the clinical threshold for admitting patients or for triaging them to critical care. This could have implied that the model could have still suffered from parameter identifiability issues when attributing severity to basic properties of the variants vs the above factors.

To reduce this risk, we conducted additional checks to ensure the model was able to reproduce the overall HFR by England region, and the age distribution of hospitalisations and deaths over time. Lastly, we also present inferred IHR, HFR and IFR by age over time and the relation in uncertainty (and overlaps) between the intrinsic R_0 and the basic severity properties of the variants by NHS England region.

Lines 175-79: “Our findings also suggest that pressures on the healthcare system had a detrimental impact on the severity of the pandemic. Between October 2020 and January 2021, There was a progressive increase in all effective severity metrics, particularly in HFR, (Figure 2A-C) as the model infers there was an overall higher risk of death in hospital, independently of the basic severity properties of the Alpha variant.” Is this a correct interpretation? When I look at figure 2B (for HFR) I see a flat line for the HFR for wildtype and alpha (separately) over this time. Put another way, the increasing HFR is a function of wildtype being replaced by alpha. Yes? (This comment also pertains to parts of the Discussion , e.g. lines 248 to 250.)

We agree with the reviewer that part of the increase in effective severity over this period is driven by the replacement of Wildtype by Alpha. However, as presented in our supplementary analysis Figure S11, there was an increase in severity that was independent of the variant replacement event. Given the absence of hospital admissions and deaths data by variant, we recognise the inherent difficulty to disentangle what proportion of this increase is due to variant vs hospital pressures. We have further clarified this in the main manuscript and supplement, as highlighted in the responses above.

468 was a progressive increase in all effective severity metrics, particularly in HFR (Figure 2A-C). The model infers there
469 was an overall higher risk of death in hospital, independently of the basic severity properties of the Alpha variant (Figure
470 S11 and Table S22). Given there was no representative data available on hospital deaths by variant during this period
471 to fit our model to, we cannot fully differentiate the specific contribution of variant and healthcare effects on the
472 increased severity. However, in an additional statistical analysis using linked patient-level records, we observed that the
473 increase in HFR during this period was positively correlated with daily critical care bed occupancy levels, with variation
474 across English regions (see Supplement section 6.1).

SUPPLEMENT

The model is a “discrete-time stochastic compartmental model”. I could not find the cycle length – is it hourly, daily, other?

Thanks for spotting this. The time step of the model is a quarter of a day, which we have now made explicit.

1.1 Model description

We adapt a previously described discrete-time (1/4 day time step) stochastic compartmental model of SARS-CoV-2 transmission (Figure S1) [1, 2, 3]. The model is an extended SEIR-type model, stratified into 17 age groups: 16 five-year age bands (0-4, 5-9, ..., 75-79) plus a group of 80+ year-olds. Mixing between age groups is informed by survey data [4].

p.5. “In contrast, the effective reproduction number, $R_{j,eff}^t$, for variant j at time t is the number of secondary infections in the actual population, accounting for immunity (natural and vaccine-induced) AND NPI POLICIES present at that time in the population.” Shouldn’t the CAPS added by me be added to this sentence?

Tony Blakely

Many thanks for the suggestion, we have made this clarification.

1.2 Reproduction number

We use two definitions of the reproduction number throughout. We denote R_t^j as the reproduction number for variant j ($j = \text{Wildtype, Alpha, Delta, Omicron}$) in the absence of immunity at time t , which varies only in response to fitted β

4

change-points (e.g. changes in non-pharmaceutical intervention (NPI) policies). This is defined as the average number of secondary infections that an individual infected at time t with variant j would generate in an entirely susceptible and unvaccinated population. In contrast, the effective reproduction number, $R_t^{j,eff}$, for variant j at time t is the number of secondary infections in the actual population, further accounting for immunity (infection- and vaccine-derived) and NPI policies present at that time in the population. Hence, by definition, $R_t^{j,eff} \leq R_t^j$.

Reviewer #2

The article "Epidemiological drivers of transmissibility and severity of SARS-CoV-2 in England" by Perez-Guzman et al. presents an in-depth analysis of the epidemiological parameters associated with different stages of the COVID-19 pandemic in England. The authors used a wide range of data sources and employed a complex mathematical model and a robust statistical framework to estimate several key parameters such as the basic reproduction number (R_0), infection hospitalization rate (IHR), infection fatality rate (IFR), and hospitalization fatality rate (HFR) for different COVID-19 variants that emerged in England.

Overall, this study provides important insights into the impact of various COVID-19 variants and highlights the importance of ongoing surveillance and monitoring of the pandemic to inform public health strategies and interventions. They found a higher IFR for the Alpha variant compared to the Delta variant. They were also able to disentangle the effects of non-pharmaceutical interventions (NPIs), immune escape, and intrinsic increase in transmissibility for the different variants of concern. They found that the Omicron variant had the higher R_0 , but the relative increase was limited with respect to the increase estimated for previous variants (due to the higher immune escape of Omicron).

I found the article very interesting as it shows how the combination of high-quality data and mathematical modelling can provide a comprehensive understanding of the factors driving the spread of the virus in England. The methods developed by the authors seems fair and robust (even if I ask the author to clarify some points, see below) and the same framework was employed by the authors in previous analyses and to inform the discussion around vaccination schedule, NPIs and reopening schedule.

Many thanks for your interest in our study and suggestions.

However, I do have a few points that might improve the quality of the paper:

A central finding in the paper is the higher IFR observed for the Alpha variant compared to the Delta VOC. This represents a novel result and in contrast with the literature. So, I think that it should be better supported.

We thank the reviewer for stressing the novelty of this result. Following this and suggestions from reviewer #1, we now actually provide estimates of the relative risk of transmissibility and severity of the variants, including the basic IFR of Alpha compared to Delta in supplement section 6.1. To obtain these, it required a slight adaptation to our approach to calculating national estimates, as we explained

above in response to reviewer #1's comments. This led to a decrease in the uncertainty of results presented.

	Wildtype	Alpha	Delta	Omicron
Wildtype	-	2.42 (2.22 - 2.61)	1.73 (1.51 - 1.96)	0.49 (0.37 - 0.60)
Alpha	0.41 (0.38 - 0.45)	-	0.72 (0.62 - 0.81)	0.20 (0.15 - 0.25)
Delta	0.58 (0.51 - 0.66)	1.40 (1.23 - 1.60)	-	0.28 (0.21 - 0.35)
Omicron	2.08 (1.66 - 2.71)	5.03 (4.01 - 6.56)	3.59 (2.89 - 4.82)	-

Table S21: Relative basic IFR risk (mean, 95%CrI) across modelled variants, calculated as IFR^j / IFR^i . Note here indices i and j refer to table row and column, respectively.

To improve this result, the analysis could include:

- a. A sensitivity analysis on changing VE for Alpha and Delta variants to see if it affects the estimated parameters, particularly IFR/HFR, in the same way as seen in the supplementary analysis for Omicron.

Many thanks for this suggestion. We have now performed these additional analyses, varying central VE parameters vs the Alpha and Delta variants by +/- 10% (Figure S8). Relatedly, we have also performed additional analyses around the choice of changepoints for the probability of death in hospital compartments, μ_D . Specifically, we explored alternative changepoints in the piece-wise linear function around the periods of emergence of the Alpha (*mu_d_winter*) and Delta (*mu_d_summer*) variants (Figure S10, last two columns).

Across all new analyses, we found only slight variations on the inferred basic severity properties of the variants, which did not impact the qualitative nature of our main findings. We would like to emphasise that, in contrast to the high number of studies on vaccine efficacy vs the Alpha and Delta variants, vaccine effectiveness vs the Omicron group of variants has been less well characterised. We thus employed robust statistical modelling to derive our pooled estimates vs the Alpha and Delta variants (supplement section 3.1) but rely on sensitivity analysis for Omicron.

End point	Dose	Central		Lower VE		Higher VE		Informed by
		AZ	PF/Mod	AZ	PF/Mod	AZ	PF/Mod	
Death	2 (Full protection)	97%	97%	80%	80%	97%	97%	Low and high values from Gov [41]
	2 (Waned protection)	56%	56%	40%	40%	70%	70%	
	3 (Full protection)	96%	96%	85%	85%	96%	96%	
	3 (Waned protection)	62%	62%	50%	50%	80%	80%	
Severe disease	2 (Full protection)	97%	97%	80%	80%	97%	97%	Low and high values from Gov [41]
	2 (Waned protection)	56%	56%	40%	40%	65%	65%	
	3 (Full protection)	96%	96%	85%	85%	96%	96%	
	3 (Waned protection)	62%	62%	40%	40%	75%	75%	
Mild disease or infection	2 (Full protection)	41%	60%	30%	30%	50%	65%	Low and high values from Gov [41]
	2 (Waned protection)	0%	0%	0%	0%	5%	15%	
	3 (Full protection)	72%	74%	60%	65%	72%	74%	
	3 (Waned protection)	0%	0%	0%	0%	20%	20%	
Transmission	2 (Full protection)	40%	40%	29%	29%	40%	40%	Assumed lower than for mild disease
	2 (Waned protection)	0%	0%	0%	0%	3%	8%	
	3 (Full protection)	40%	40%	30%	30%	50%	50%	
	3 (Waned protection)	0%	0%	0%	0%	10%	10%	

Table S16: Sensitivity analysis parameters for vaccine effectiveness against the Omicron variant for AstraZeneca (AZ), Pfizer (PF), and Moderna (Mod) by vaccine dose. "Infection" refers to vaccine effectiveness protecting an individual from being infected with SARS-CoV-2, whilst "transmission" refers to the vaccine effectiveness at preventing onward transmission by an infected individual.

Figure S8: Sensitivity analysis of model inferred intrinsic R_0 and basic IHR, HFR and IFR of the variants. From left to right: Central, high and low vaccine efficacy (1st and 2nd dose, table S14) against the Alpha variant, high and low vaccine efficacy (1st, 2nd and booster, table S17) against the Delta variant, and high and low vaccine efficacy (2nd and booster dose only table S18) against the Omicron variant. Box plots show mean model-inferred properties and 95% CrI for the Wildtype (grey), Alpha (blue), Delta (orange) and Omicron (pink) variants.

5.3 Other sensitivity analysis

Our model allowed fitting time-varying changes in severity, independent of the inferred basic properties of the variants. We explored alternative parameterisations of the piece-wise linear function for changing $\mu_D(t)$ (see details in section 4.4.4). In these scenarios, we specifically sought to infer whether the choice of change points influenced the inferred relative severity between the Alpha and Delta variants. Additionally, we varied our central serial interval parameters, by either fixing it at its highest (i.e. all variants with same value as Wildtype) and lowest (i.e. all variants with same value as Omicron BA.1) values.

Scenario	Dates	Rationale
Central	01-04-2020	See table S10
	01-07-2020 to 15-09-2020	
	15-10-2020 to 01-12-2020	
	04-02-2021	
	01-04-2021 to 04-11-2021	
$\mu_D(t)_{winter}$	31-12-2021	Explore a linear change in $\mu_D(t)$ from 09-2020 to 12-2020 to infer potential changes in severity given mounting healthcare demands around the date of Alpha emergence.
	01-04-2020	
	01-07-2020 to 15-09-2020	
	01-12-2020	
	04-02-2021	
$\mu_D(t)_{summer}$	01-04-2021 to 04-11-2021	Explore flat period in $\mu_D(t)$ between 04-2021 and 06-2021 during the period of Delta emergence, then linear change from 06-2021 to 11-2021 as Delta was dominant during a period of potential evolving hospital admission and triaging thresholds.
	31-12-2021	
	01-04-2020	
	01-07-2020 to 15-09-2020	
	15-10-2020 to 01-12-2020	
$\mu_D(t)_{summer}$	04-02-2021	Explore flat period in $\mu_D(t)$ between 04-2021 and 06-2021 during the period of Delta emergence, then linear change from 06-2021 to 11-2021 as Delta was dominant during a period of potential evolving hospital admission and triaging thresholds.
	01-04-2021 to 01-06-2021	
	04-11-2021	
	31-12-2021	
	01-04-2020	

Table S17: Sensitivity analysis of fitted change points for the time-varying $\mu_D(t)$.

Figure S10: Sensitivity analysis of model inferred intrinsic R_0 and basic IHR, HFR and IFR of the variants. From left to right: Central, high and low fixed serial interval duration (see values in table S13), and alternative changepoints for $\mu_D(t)$ in the winter of 2020/21 and in the summer of 2021 (see values and rationale in table S17). Box plots show mean model-inferred properties and 95% CrI for the Wildtype (grey), Alpha (blue), Delta (orange) and Omicron (pink) variants.

- b. *In the discussion, the author mentions the role of AY.4.2 sub-variant in reducing the severity compared to non-AY.4.2 ones. It would be useful, if the authors have data (or if they have ways to separate the two periods), to estimate the different IFR when variant AY.4.2 is non-dominant/dominant to see if the estimates in this case align with literature (reporting higher IFR for Delta compared to alpha during the first-time window). This analysis could be included in the SI.*

Many thanks for highlighting this. Delta AY.4.2 never became dominant in England, peaking at <30% frequency in daily PCR-confirmed cases from the community in late 2021.¹ After this point both Delta variants were rapidly and completely replaced by Omicron BA.1. We were thus unable to assess its specific contribution on changing patterns of severity, as our framework relies on a complete or near-complete dominance to robustly infer intrinsic and basic properties for a variant. We have made a clarification on this in our discussion.

545 driven severity characteristics over the entire period of variant circulation. As a result, their estimates will be subject to
546 biases inherent to the profile of patients seeking a PCR test in the population during the early phase of Delta emergence,
547 which would not be representative of the profile of PCR-seeking behaviours across the whole period of Alpha and Delta
548 transmission. Additionally, the period of Delta dominance was characterised by a partial replacement of the original
549 Delta variant by the AY.4.2 sub-variant,³⁴ which showed a decreased severity.³⁵ Whilst we did not explicitly model
550 Delta AY.4.2, as it never completely replaced the original Delta variant in England,³⁴ our inferred severity estimates for
551 Delta include both periods of original Delta and AY.4.2 variant circulation.

- 2) *The second main finding is related to the parameters assessed for the Omicron variant. The authors discuss this outcome in the discussion section, stating “Our analysis robustly shows that the initial (BA.1) sub-variant of this lineage has basic severity properties comparable to Wildtype IHR, HFR and IFR “. However, based on Figures 2A-C, it appears that Omicron has a basic IHR that is approximately 60% higher than that of the Wildtype, whereas its basic HFR and IFR are lower by approximately 70% and 60%, respectively. Therefore, I find this sentence incorrect and I would advise the authors to revise it or clarify why they consider those values for Omicron comparable to Wildtype.*

Thank you for raising this issue. We agree that the statement as presented was inaccurate and have amended it to only discuss basic severity properties. We had intended to discuss that the inferred basic severity of Omicron is comparable to the effective severity of the pandemic in late 2020, at a point when Wildtype was still dominating.

655 Currently, the Omicron variant lineage dominates the pandemic landscape globally.²⁵ Our analysis robustly shows that
656 the initial (BA.1) sub-variant of this lineage has a basic IHR comparable to other variants. This highlights that, albeit
657 Omicron’s basic HFR and IFR are substantially lower than other variants, the Omicron lineage of SARS-CoV-2 variants
658 remain a public health threat. Recent analyses from China, where cumulative levels of protection against the virus in

- 3) *On a technical note, the authors employ a two-step process for model fitting that involves using a classical MCMC procedure to explore the parameter space for pre-tuning, followed by a second step with pMCMC using 192 particles (as described in section 1.3 of the SI). I would like to understand the benefits of the pMCMC stage. Traditional MCMC is easier to implement and can be more easily reproduced by other researchers that have similar data with no need to have two steps. Could the authors clarify the benefits of using pMCMC over the traditional MCMC alone? I understand that this could be useful for real-time responses to speed up the inference process,*

but since this is a retrospective study I'm not sure about the benefits. The figures S28-32 show the priors before the traditional MCMC step and the posterior at the end of the inference. I am curious about the posterior distribution parameters of the MCMC step that are used as initial values and proposal kernel for the pMCMC. Is it possible for the authors to superimpose the posterior distribution parameters onto these figures? Could the authors provide additional information about the MCMC procedure used, such as the type of algorithm, number of iterations, whether different starting conditions were tested, and how convergence has been checked?

Additionally, I am wondering why the authors used only 192 particles to estimate such a large number of parameters. I would have expected a much larger number of given the number of parameters that they are estimating. They should also clarify if they have performed sanity checks on that (if results are similar with a higher number of particles).

Many thanks for suggesting this clarification. For clarity we break down our response into three relevant topics, which we have now also summarised more in Supplementary sections 4.11 (running the model), 6.5 (number of particles check) and 6.9 (assessing convergence):

- **Stochastic vs deterministic modelling:** most transmission models developed during the pandemic, particularly for real time analyses, have been deterministic. They are easier to fit to data using, for example, traditional MCMC methods. However, stochastic models excel at capturing uncertainty in processes involving small numbers and can absorb structural uncertainty. This was particularly relevant for some of the specific aims of our study, such as capturing the emergence of variants of concern, among other epidemiological events with small counts (e.g. hospitalisations and deaths by age groups). Stochastic models need more involved fitting procedures, for which *pMCMC* has been postulated, as it uses sequential Monte Carlo (SMC) to avoid the need to data augmentation techniques (i.e. as would be required by traditional MCMC). That is, SMC uses Monte Carlo estimates of latent variables (unobserved states) to avoid the need to estimate them alongside parameter inference (MCMC). This was also crucial for our modelling framework, as we explicitly account for a large number of unobserved states.
- **Two step fitting procedure:** *pMCMC* is more challenging to fit and can be slow to converge, compared to MCMC, given each step involves the simulation of many particles. We therefore used an “expectation” (deterministic) equivalent of our model, which instead of random draws uses the mean of the corresponding distribution (e.g. $n * p$ for a binomial draw) coupled with an adaptive MCMC. This allowed us to accelerate convergence and 85-dimensional parameter space exploration, which we deem as a “warm-up” step using an emulator. Supplementary table S24 and figures S37 to S43 summarise our convergence

diagnostic tools, which included visual inspection of MCMC chains, Gelman-Rubin statistics and Effective Sample Size.

- **Number of particles:** as stated by the reviewer, 192 particles could be deemed as a small number of particles given our multi-dimensional parameter space. A problem with pMCMC chains is they can easily become “sticky” (i.e. do not move),² due to evaluations of marginal likelihoods that are small compared to their actual value (due to the variance of the estimator). To avoid this, we “regenerated” our likelihood estimation with a probability 1/100 (i.e. re-estimate the likelihood with the same parameters). This is a common technique in pMCMC, which while introducing a small bias, performs well in practice.^{2,3} We, nevertheless, performed an additional analysis with twice the number of particles, at 384, as summarised in supplementary section 6.5. This came at an approximately double running time in computational cost and, of note, results around inferred intrinsic transmissibility and basic severity of the variants remained similar.

4.11 Running the model

The model is fitted to multiple data streams up to 24th February 2022, capturing the entirety of the SARS-CoV-2 epidemic up to the official end of the policy for self-isolation in England [72]. The model is run under baseline assumptions reflected in our fixed (Table S11) and VE parameters (Table S4).

Before running the pMCMC, we pre-tune the model by running a traditional MCMC on the equivalent “expectation model”, defined as the same model but wherever a random draw arises, the mean of the corresponding distribution is used instead, thereby allowing compartments to take non-integer values. This was done to optimise computational efficiency in exploring the multi-dimensional parameter space across fitted parameters (figure S33, figure S34, figure S35 and figure S36). The “expectation” model is coupled with an adaptive MCMC to accelerate parameter space exploration.

We then use the pre-tuned parameter set from the “expectation” model with the highest posterior and variance-covariance (VCOV) matrix of the posterior distribution parameters as the initial values and proposal kernel, respectively, for subsequent pMCMC runs. The latter uses sequential Monte Carlo estimates of the latent variables with a bootstrap particle filter. This avoided both the need for data augmentation techniques and to estimate latent variables (i.e. unobserved states) alongside fitted parameters.

At each iteration of the pMCMC, we randomly rerun the particle filter on the current parameter set with probability $\frac{1}{100}$ to get a new marginal likelihood estimate, which prevents chains from getting stuck at a particular parameter due to a high likelihood estimate [73, 74]. We run 4 chains with 192 particles in this process over 5,000 pMCMC iterations, of which 1000 are discarded as burn-in. We thin the combined sample uniformly to achieve a posterior sample size of 1000. To test the robustness of our choice of particle number, we also ran our model with double the number of particles (384) for our central assumptions (table S23), which comes at approximately double the run-time cost. Given results were generally similar, we are confident that using 192 particles consistently for our central assumption fits and all sensitivity analyses provides robust results in a more reasonable run-time.

6.5 Number of particles check

To check on the robustness of our results given selection of number of particles for the pMCMC, we compare inferred values of the intrinsic R_0 and basic severity with central parameters using 192 (default) and 384 (double) pMCMC particles. We ran these from the same initial (pre-tuned) parameter values and VCV proposal matrix (see section 4.1).

	Intrinsic R_0	Basic IHR	Basic HFR	Basic IFR
192 particles (default)				
Wildtype	2.6 (2.4-2.7)	2.2 (2.1-2.3)	32.6 (30.7-34.4)	1.2 (1.1-1.2)
Alpha	4.2 (3.9-4.4)	3.3 (3.0-3.5)	48.5 (44.3-52.9)	2.9 (2.7-3.2)
Delta	7.0 (6.5-7.4)	4.2 (4.1-4.4)	32.6 (29.2-36.6)	2.2 (2.0-2.4)
Omicron	8.3 (7.7-8.8)	3.4 (3.1-3.8)	14.3 (11.3-17.1)	0.7 (0.6-0.8)
384 particles				
Wildtype	2.5 (2.3-2.6)	2.3 (2.2-2.4)	31.0 (29.4-32.4)	1.1 (1.0-1.2)
Alpha	4.0 (3.8-4.2)	3.5 (3.4-3.7)	46.4 (43.2-49.5)	2.7 (2.5-2.9)
Delta	6.8 (6.5-7.2)	4.4 (4.2-4.5)	29.0 (26.0-31.2)	1.9 (1.7-2.1)
Omicron	8.1 (7.8-8.5)	3.5 (3.2-3.7)	12.9 (9.9-15.4)	0.6 (0.5-0.7)

Table S23: Main model results (see in main manuscript) with default of 192 pMCMC particles vs 384.

6.9 Assessing convergence

As previously described (section 4.1), our framework followed a two-step fitting process whereby we first ran eight parameter pre-tuning MCMC chains ("expectation" model), followed by four pMCMC chains (stochastic). Note both the "expectation" and stochastic steps are the same model and share the same parameter space.

Each of the eight MCMC chains were started from different places of the 85-dimensional parameter space. To assess convergence of the parameter samples produced by the MCMC chains we performed:

- Visual inspection of the MCMC chains (Figures S37 to S43). The caterpillar-like appearance of the chains (8) for each parameters plus the shape of the log-likelihood curve suggests convergence and good coverage of the target area of the parameter space.
- Gelman-Rubin statistic (Table S24) associated with the full parameter chains (30,000 MCMC steps). All chains show \hat{R} statistics below the recommended 1.1 threshold.
- Effective Sample Size (ESS) for the different parameters (Table S24) for the full parameter chains (30,000 MCMC steps).

Indeed, the above diagnostics suggest the MCMC chains reached the right area of the parameter space and that, therefore, the scales of the proposal distribution for the random walk were appropriate for the following pMCMC algorithm. That is, MCMC chains reduced auto-correlation between successive samples. Subsequently, we used a re-scaled version of the VCV proposal matrix generated by the MCMC in the pMCMC algorithm. This was done in order to capture the main correlation structure and the "scale" of the different parameters, where diagonal elements of the VCV matrix represent the variances of the different parameters.

It should be noted that there are no existing diagnostics to guarantee MCMC convergence, but only necessary tests. Nevertheless, by relying on a group of diagnostics rather than a single one, we thoroughly tested the validity of our fitting process, as at least one of these diagnostics would have failed otherwise. Please see the posterior distributions of the fitted parameters after the pMCMC in figures S34 to S38. Where informative priors were used, their distributions are shown (red lines).

Table S24: Gelman-Rubin and effective sample size (ESS) of pre-tuned model parameters after MCMC ("expectation" model). For each NHS England region, we present the highest Gelman-Rubin and lowest ESS values across the 85-dimensional parameter space. See corresponding convergence plots for each parameter and region below (figure S37 to figure S43).

Diagnostic	East of England	London	Midlands	North East and Yorkshire	North West	South East	South West
Gelman – Rubin	1.09	1.09	1.05	1.06	1.08	1.08	1.09
ESS	1433	1412	1479	1282	1475	1214	1272

Minor Comments:

- Table S4 indicates that the protection against transmission for the Omicron variant is set at 50% for individuals who have just received their third dose, which is higher than the protection against Delta variant for the same people. However, the protection against transmission provided by the second dose against Omicron is either equal (Full protection) to or lower (Waned protection) to the one against Delta. What is the rationale behind this decision?

Thanks for highlighting this. We had made a mistake regarding the booster VE value against transmissibility by the Omicron variant. Given the lack of data to support an empiric estimation, as we did for the corresponding Alpha and Delta values, we had intended to assume the same value as for Delta, at 40%. We made this correction and reran all analyses. The conclusions and results remain qualitatively and quantitatively unchanged.

- Ref 68 of the SI is incomplete.

Many thanks for highlighting. We have corrected this reference by Chen et al.

- Why in figure 2A-C in the effective IHR/HFR/IFR the line of wildtype is starting just before October 2020 and not from the beginning (March 2020)?

Many thanks for highlighting this. This was a labelling issue in our plotting code, which we have now amended.

482

483

Figure 2 – Inferred severity of SARS-CoV-2 variants in England between March 2020 and February 2022. A-C)

- *Table S9: in the last two rows of the table the Rationale reported is “something”.*

Apologies for this oversight. We now realise we had not updated this table appropriately prior to submission. The SI was developed in *LaTeX*, with placeholder tables pre-populated with “*Something*” before editing. These two dates correspond to the timing of rollout of novel anti-Covid oral therapies aimed at individuals at high risk of severe disease outcomes. Additionally, we had not updated the definitive dates used for $\mu_D(t)$ in our code as central model values. This is also now correct. Please note that, as presented in response and snapshots above, we performed a new sensitivity analysis, exploring alternative change points for $\mu_D(t)$, which we find had no effect on the magnitude or qualitative nature of the relative basic severity of the variants, in particular of Alpha compared to Delta.

Parameter	Dates	Rationale	Reference
$\mu_H(t)$	04-11-2021	Approval and roll-out of novel outpatient treatments for COVID-19.	[51, 52, 53, 54]
	31-12-2021		
$h_{ICU}(t)$	01-04-2020	First hospital treatment protocols and use of dexamethasone established.	[55, 56]
	01-06-2020		
$h_{GD}(t)$	01-05-2020	Potential change in healthcare seeking behaviour or case management after the first wave in the community/carehomes.	NA
	01-07-2020		
$\mu_D(t)$	01-04-2020	First wave	NA
	01-07-2020 to 15-09-2020	Trough after first wave	NA
	15-10-2020 to 01-12-2020	Winter 2020/21 wave	NA
	04-02-2021	Trough after winter 2020/21 wave	NA
	01-04-2021 to 04-11-2021	Approval and roll-out of novel	
	31-12-2021	oral treatments for COVID-19	[51, 52, 53, 54]

Table S10: Fitted changepoints for time-varying severity parameters with piecewise form. We explored alternative change points for $\mu_D(t)$ in sensitivity analysis (see section 5.3).

References

1. UK Health Security Agency. *SARS-CoV-2 variants of concern and variants under investigation in England Technical briefing 49*.
https://assets.publishing.service.gov.uk/government/uploads/system/uploads/attachment_data/file/1129169/variant-technical-briefing-49-11-january-2023.pdf (2023).
2. Andrieu, C., Doucet, A. & Holenstein, R. Particle Markov Chain Monte Carlo Methods. *J. R. Stat. Soc. Ser. B Stat. Methodol.* **72**, 269–342 (2010).
3. Andrieu, C. & Roberts, G. O. The pseudo-marginal approach for efficient Monte Carlo computations. *Ann. Stat.* **37**, (2009).

REVIEWERS' COMMENTS

Reviewer #2 (Remarks to the Author):

In the reviewed version of the article "Epidemiological drivers of transmissibility and severity of SARS-CoV-2 in England" by Perez-Guzman et al, the authors have addressed all of my comments and concerns. I have no additional feedback to provide at this stage.